# A Closer Look at Offline RL Agents

**Yuwei Fu,  Di Wu,  Benoit Boulet**
McGill University
yuwei.fu@mail.mail.ca, {di.wu5, benoit.boulet}@mcgill.ca

## Abstract

Despite recent advances in the field of Offline Reinforcement Learning (RL), less attention has been paid to understanding the behaviors of learned RL agents. As a result, there remain some gaps in our understandings, *i.e.*, why is one offline RL agent more performant than another? In this work, we first introduce a set of experiments to evaluate offline RL agents, focusing on three fundamental aspects: representations, value functions and policies. Counterintuitively, we show that a more performant offline RL agent can learn relatively *low-quality* representations and *inaccurate* value functions. Furthermore, we demonstrate that the proposed experiment setups can be effectively used to diagnose the bottleneck of offline RL agents. Inspired by the evaluation results, a novel offline RL algorithm is proposed by a simple modification of IQL and achieves SOTA performance. Finally, we investigate when a learned dynamics model is helpful to model-free offline RL agents, and introduce an uncertainty-based sample selection method to mitigate the problem of model noises. Code is available at: https://github.com/fuyw/RIQL.

## 1 Introduction

Offline Reinforcement Learning (RL), also known as batch RL, refers to the problem of learning effective control policies from a fixed offline dataset [39]. Due to the wide availability of logged-data and the increasing computing power, offline RL holds the promise for successful real-world applications [40]. For example, offline RL suits well for scenarios where collecting online data is time-consuming, dangerous or unethical, *i.e.*, robotics, self-driving cars and medical treatments [21]. While most off-policy RL algorithms are applicable in the offline setting, they usually suffer from the extrapolation error [18, 35, 11] due to out-of-distribution (OOD) samples. Different solutions have been proposed to mitigate this problem, *i.e.*, adding constraints [18, 50], behavior cloning (BC) [7, 57], learning a dynamics model [55, 28, 3], incorporating uncertainties [51], using ensembles [1], or learning pessimistic value functions [36, 5, 26].

Recently, offline RL algorithms have shown to be effective to solve various challenging tasks [46, 42]. However, most of these works mainly focus on designing new algorithms and less attention has been paid to understanding the behaviors of the learned offline RL agents. As a result, some basic questions are still poorly understood. For example:

- Why does one offline RL agent perform better than other baseline agents in a benchmark task, as illustrated in Fig 1?
- Does the more performant agent learn better representations or more accurate value functions?
- Which of the existing offline policy evaluation/improvement methods is more effective?
- When is a learned dynamics model helpful to a model-free offline RL agent?
- Given a learned dynamics model, how can we use the generated samples more efficiently?

36th Conference on Neural Information Processing Systems (NeurIPS 2022).

| Baseline | Paradigm | Constraint Type | Speed | Disadvantage |
|----------|----------|-----------------|-------|--------------|
| TD3+BC [16] | Policy-regularized | Actor | ✓ | Fail in difficult tasks |
| CQL [36] | Pessimism | Critic | ✗ | Low computation efficiency |
| COMBO [54] | Model-based | Critic | ✗ | Sensitive to model noises |
| IQL [30] | Generalized BC | Both | ✓ | Need to tune parameter $\tau$ |

Table 1: A brief summary of four SOTA offline RL baselines from different paradigms.

In spite of several potential intuitions, there is a lack of clear explanations for these questions. Such incomplete understanding impedes progress, as researchers fail to identify the full impact of algorithmic changes without extensive tuning [12]. Given the increasing number of newly proposed offline RL algorithms, we argue that it is time to benchmark current SOTA offline RL algorithms [15] and shed some lights on these fundamental questions. To this end, we empirically compare four SOTA offline RL baseline algorithms, as shown in Table 1, from different paradigms on the standard D4RL dataset [14].

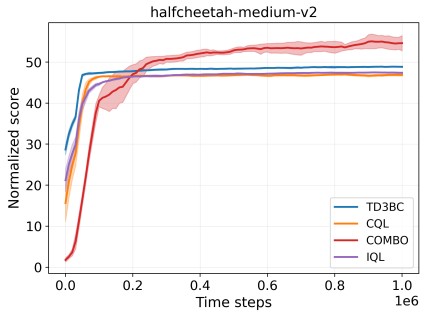

Figure 1: Why does the COMBO agent perform better than other baselines?

In this work, we design a comprehensive set of experiments to compare the *representations, value functions and policies* of offline RL agents in order to gain a deeper understanding of the behaviors of SOTA offline RL agents. We identify a surprising discovery that a more performant agent sometimes has *worse* representations and *inaccurate* value functions. This motivates us to take a closer look at the learned policies. Our empirical results show that a performant offline RL agent is usually able to select better sub-optimal actions while avoiding bad ones. Furthermore, we demonstrate that the proposed experiment setups can be used to evaluate the effectiveness of existing policy evaluation/improvement methods. As a case study, we introduce a variant of IQL [30] by relaxing the in-sample constraint for the policy improvement step, which achieves better performance. Moreover, we investigate when a learned dynamics model can help a model-free offline RL agent, and we propose an uncertainty-based sample selection method to mitigate the problem of model noises. Our contributions are as follows:

- We conduct extensive experiments, focusing on representations, value functions and policies, to compare different SOTA offline RL agents and explain some fundamental questions.

- We show the effectiveness of the proposed experiment setups in diagnosing the bottleneck of offline RL agents with a new variant of the IQL algorithm that achieves SOTA performance.

- We investigate when a learned dynamics model helps model-free offline RL agents, and we introduce an uncertainty-based sample selection method that is more robust to model noises.

## 2 Background

A Markov Decision Process (MDP) $\mathcal{M} = \langle \mathcal{S}, \mathcal{A}, \mathcal{R}, \mathcal{P}, \gamma \rangle$ [44] is specified by a state space $\mathcal{S}$, an action space $\mathcal{A}$, a transition kernel $\mathcal{P} : \mathcal{S} \times \mathcal{A} \to \Delta(\mathcal{S})$, a reward function $\mathcal{R} : \mathcal{S} \times A \to \mathbb{R}$, and a discount factor $\gamma \in [0, 1)$. The goal is to find a policy $\pi(a|s) : \mathcal{S} \to \Delta(\mathcal{A})$, which maps from state to distribution over actions, that maximizes the expected cumulative discounted reward $J(\pi) := \mathbb{E}_\pi[\sum_{t=0}^{\infty} \gamma^t r_t]$. The performance of the policy can be defined by the value functions $Q^\pi(s, a) := \mathbb{E}_\pi [\sum_{t=0}^{\infty} \gamma^t r_t | s_0 = s, a_0 = a]$ and $V^\pi(s) := \mathbb{E}_\pi [\sum_{t=0}^{\infty} \gamma^t r_t | s_0 = s]$, where $\mathbb{E}_\pi[\cdot]$ is the expected result when following the policy $\pi$.

In deep RL, $Q$-function is parameterized with a neural network $Q_\theta^\pi(s, a)$. Following prior works [20, 34, 32, 38], we denote the penultimate layer of the neural network as the learned *representation*, a $d$-dimensional mapping $\phi(s, a) : \mathcal{S} \times \mathcal{A} \to \mathbb{R}^d$. Such that the $Q$-function is linear in the representation $Q_\theta^\pi(s, a) = \theta^\top \phi(s, a)$, where $\theta \in \mathbb{R}^d$ is a vector of weights. If the state space $\mathcal{S}$ and action space $\mathcal{A}$ are finite, then the representation corresponds to a feature matrix $\Phi \in \mathbb{R}^{|\mathcal{S}| \cdot |\mathcal{A}| \times d}$, whose rows are the vector $\phi(s_i, a_i) \in \mathbb{R}^d$ for the $i$-th state-action pair $(s_i, a_i)$.

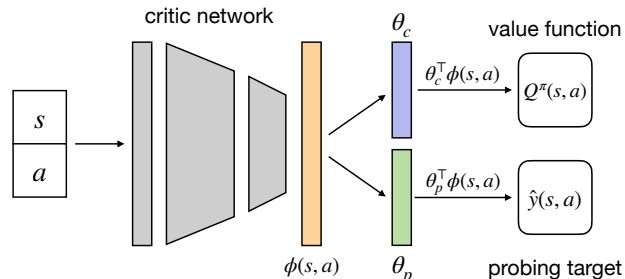

Figure 2: An illustration of the *representation probing* experiment.

| Inductive Bias | Target | Probing Function |
|---|---|---|
| Transition Dynamics | reward $r$ | $r = f_p(\phi(s,a))$ |
| | next state $s'$ | $s' = f_p(\phi(s,a))$ |
| | inverse action $a$ | $a = f_p(\psi(s), \psi(s'))$ |
| Optimal Policy | optimal action $a^*$ | $a^* = f_p(\psi(s))$ |
| | optimal $Q^*(s,a)$ | $Q^*(s,a) = f_p(\phi(s,a))$ |
| | optimal $V^*(s)$ | $V^*(s) = f_p(\psi(s))$ |

Table 2: Different probing targets.

In this work, we study the offline RL setting [40], where we aim to learn a policy $\pi(a|s)$ purely from a fixed offline dataset $\mathcal{D} = \{(s_i, a_i, s_i', r_i)\}$, generated from a behavior policy $\pi_\beta(a|s)$. Most of recent offline RL algorithms build on the *Approximate Dynamics Programming* (ADP) method [4] that learn the $Q_\theta^\pi(s,a)$ by minimizing the temporal difference error:

$$L_{TD}(\mathcal{D}, \theta) = \mathbb{E}_{(s,a,r,s') \sim \mathcal{D}} \left[ (r + \gamma \max_{a'} Q_{\hat{\theta}}^\pi(s', a') - Q_\theta^\pi(s,a))^2 \right], \qquad (1)$$

where $Q_{\hat{\theta}}^\pi(s,a)$ is the target network. A major challenge in offline RL is the issue of distributional shift between the learned policy $\pi(a|s)$ and the behavior policy $\pi_\beta(a|s)$. Specifically, the OOD actions $a'$ can produce erroneously over-estimated target values for $Q_{\hat{\theta}}^\pi(s', a')$ in Eq 1. Therefore, many existing offline RL algorithms are motivated to constrain the learned policy to stay close to the behavior policy [50, 16], or penalize large over-estimated $Q$-values [36, 11]. We provide more extensive backgrounds in Appendix A.

## 3 Representation evaluation experiments

Inspired by the huge success achieved by representation learning in (un)supervised learning [6, 25, 45], it is natural to ask – does a more performant offline RL agent learn better representations? To answer this question, we first leverage the *representation probing* technique [2, 23, 37] to evaluate the learned representations of each baseline agent. Then we use some latest introduced *representation metrics* [34, 33, 38, 41] as proxies to evaluate the quality of the learned representations.

### 3.1 Representation probing experiment

For a state-action pair $(s, a)$, we use the output of the penultimate layer of the critic network as the critic representation $\phi(s,a) \in \mathbb{R}^d$, as shown in Fig 2. We later train another linear model $f_p(\cdot)$ to predict a probing target, such as reward $r$, by linear regression $\hat{y}(s,a) = f_p(\phi(s,a)) = \theta_p^\top \phi(s,a)$. We evaluate the actor embedding $\psi(s) \in \mathbb{R}^d$ in a similar way. To validate whether the learned representations contain meaningful inductive biases [48, 53], we select two categories of probing targets (Table 2). We first use the next state, reward, and action to detect if the representations learned any semantics about the *transition dynamics*. We then directly use the optimal action and optimal value functions to check if the representations learned any information about the *optimal policy*.

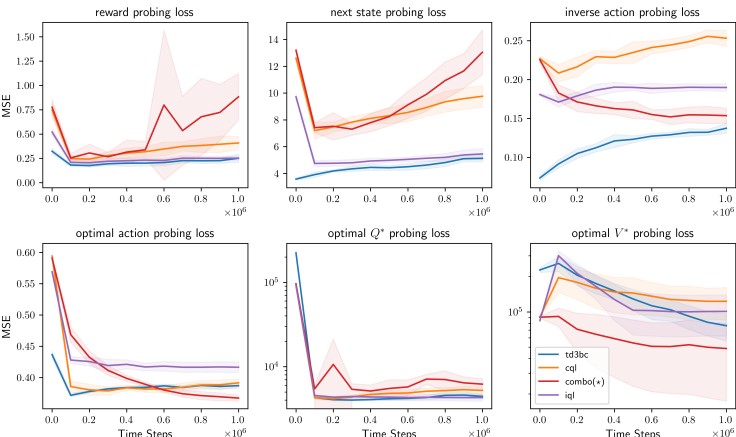

Figure 3: Representation probing experiment result on the *halfcheetah-medium-v2* environment. We label the most performant agent with a ($\star$) mark. Curves are averaged over 5 seeds.

|  | reward | next state | inverse action | optimal action | optimal $Q^*$ | optimal $V^*$ |
|---|---|---|---|---|---|---|
| #Env | 0 | 0 | 3 | 5 | 4 | 4 |

Table 3: Number of environments that the most performant agent has the least probing loss.

In the experiment, an online TD3 agent [17] is used to approximate the optimal policy $\pi^*$. We use five checkpoints of the online agent (with different levels of performance) to collect 100K transitions as the probing dataset $\mathcal{D}_{probe}$. For each probing target, we use a 5-fold cross-validation on $\mathcal{D}_{probe}$ to train a linear regression model with Mean Squared Error (MSE) loss. The result of the probing experiment on *halfcheetah-medium-v2* environment is shown in Fig 3. Results on other environments are summarized in Appendix C.1. We also list the number of environments where the most performant offline RL agent has the least probing loss in Table 3.

We can observe that the most performant agent usually has worse probing results except for the *optimal action* experiment. These results indicate that the transition dynamics-based information is not that important for an offline RL agent to perform well in the selected benchmark tasks. Further, many baseline agents suffer from large MSE losses on the *optimal value functions* $Q^*(s,a)$ and $V^*(s)$ experiments, which highlights the difficulty to learn accurate value functions in such offline setting due to the limited data coverage and additional policy/value constraints. In addition, as long as the actor representation $\psi(s)$ preserves the ability to learn good actions (low optimal action probing loss), then the offline RL agent holds the potential to achieve good performance.

## 3.2 Representation metric experiment

We further utilize some recently proposed *metrics* [34, 33] as proxies to evaluate the quality of the learned representations. We start with the following definitions and more details are in Appendix A.2.

**Definition 1** (Feature dot-product). *Feature dot-product $\phi(s,a)^\top \phi(s',a')$ is the dot-product of two critic representations [34], where $s' \sim \mathcal{P}(\cdot|s,a)$ and $a' \sim \pi(\cdot|s')$ is the next state and next action.*

**Definition 2** (Effective rank). *Effective rank $\mathrm{srank}_\delta(\Phi) = \min\left\{k : \frac{\sum_{i=1}^k \sigma_i(\Phi)}{\sum_{i=1}^d \sigma_i(\Phi)} \geq 1 - \delta\right\}$ of a feature matrix $\Phi \in \mathbb{R}^{|\mathcal{S}| \cdot |\mathcal{A}| \times d}$ approximates the rank of $\Phi$ [33], where $\{\sigma_i(\Phi)\}$ are the singular values of $\Phi$ in the decreasing order ($\sigma_1 \geq \cdots \geq \sigma_d \geq 0$) and $\delta$ is a threshold parameter, i.e., $0.01$.*

**Definition 3** (Effective dimension). *Effective dimension $d_{\mathrm{eff}}(\Phi) = N \max_{i=1,\cdots,N} \|P_\Phi e_i\|_2^2$ of a feature matrix $\Phi \in \mathbb{R}^{|\mathcal{S}| \cdot |\mathcal{A}| \times d}$ measures the sparsity of the column space of $\Phi$ [38], where $N = |\mathcal{S}| \cdot |\mathcal{A}|$ and $P_\Phi$ is the orthogonal projector onto the column space of $\Phi$.*

For the *feature dot-product* metric, we also compute the *cosine similarity* $\frac{\phi(s,a)^\top \phi(s',a')}{\|\phi(s,a)\|\|\phi(s',a')\|}$ to decouple the effect the representation norm. To compute the *effective rank*, we first compute the critic

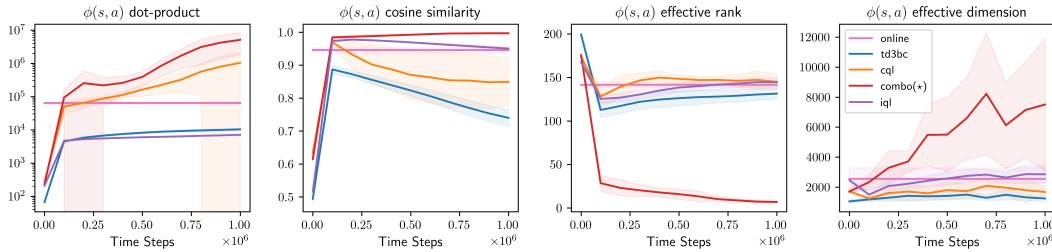

Figure 4: The best performing COMBO agent learns *low-quality* representations with large norms, in which $\phi(s, a)$ is very close to $\phi(s', a')$. Moreover, feature space collapses significantly.

representation $z_i = \phi(s_i, a_i)$ for each sample in the probing dataset $\mathcal{D}_{probe}$. Then we approximate the covariance matrix of $\Phi$ by $C(\Phi) = \frac{1}{|\mathcal{D}_{probe}|} \sum_i (z_i - \bar{z})(z_i - \bar{z})$, and use SVD on $C(\Phi)$ to compute the singular values $\{\sigma_i(\Phi)\}$ [27]. Unlike the original implementation of the *effective dimension* which used a fixed threshold to approximate the rank of $\Phi$, we instead use the *effective rank* $\mathrm{srank}_\delta(\Phi)$. More details are discussed in Appendix C.2.

Fig 4 shows the experiment results on the *halfcheetah-medium-v2* environment. Results on other environments are summarized in Appendix C.2. We also add the result for the online TD3 agent for reference. We can observe that the most performant COMBO agent has most severe *feature co-adaptation* [34] (large feature dot-product) and *representation collapse* [33] (small effective rank) problems, which indicate it learned pretty "low quality" critic representations. In terms of these metrics alone, some baselines agents such as TD3+BC and IQL seem to learn similar or better critic representations than the *near-optimal* online agent. These results are consistent with our findings in the previous probing experiments, which indicate that a performant offline RL agent sometimes does not necessarily need high quality critic representations.

> **Observation 1.** A performant offline RL agent sometimes learn low-quality representations.

## 4 Value ranking experiments

As described in the last section, a performant offline RL agent can learn low-quality representations. We therefore turn our focus to the value functions to investigate that – does a performant offline RL agent learn more accurate $Q$-functions? Since the pessimism-based offline RL algorithms adopt an additional penalty to learn lower-bounded $Q$-functions [36, 54], it could be inappropriate to measure the accuracy of $Q$-functions using metrics like MSE. Therefore, we design the following *value ranking experiments* that focus on the ability of $Q$-functions to rank actions. The motivation is simple – a "good" $Q$-function in offline RL can have large MSE loss, but it should be good at ranking actions, such that $Q^\pi(s, a_i) > Q^\pi(s, a_j)$ if $Q^*(s, a_i) > Q^*(s, a_j)$ where $Q^*$ is the optimal $Q$-function.

In the experiment, we first use different behavior policies (four baseline agents and a near-optimal online agent) to interact with the environment to collect a validation set $\mathcal{D}_{value} = \{s_1, \cdots, s_N\}$. At each state $s_i$, we use each baseline agent to sample $m$ different actions, *i.e.*, $\pi(\cdot|s_i) + \epsilon$ where $\epsilon$ is a Gaussian noise. Therefore, we have $M = 4m$ different actions $A_i = \{a_{i1}, \cdots, a_{iM}\}$ for each state. Then we use baseline agents to rank the $M$ actions at state $s_i$, *i.e.*, $R^{\pi_j}(s_i) = \mathrm{sort}(Q^{\pi_j}(s_i, a_{i1}), \cdots, Q^{\pi_j}(s_i, a_{iM}))$ where $Q^{\pi_j}$ is the learned $Q$-function of the $j$-th baseline agent. We use two metrics to evaluate the accuracy of the learned $Q$-functions: (1) Spearman's rank correlation coefficient (Rank IC) and (2) TopN accuracy. We approximate the optimal rank information using the online agent. Here, the larger rank IC/topN accuracy is, the more accurate is the $Q^\pi(s, a)$ at ranking actions. More details are described in Appendix D.

Experiment results are shown in Table 4 and Table 5, where we report the mean result and standard deviation across 5 random seeds. The most performant baseline agents in each environment are colored brown. We can observe that many best-performing agents have near-zero rank IC and lower TopN accuracy results. This indicates that the learned $Q$-functions are less capable of distinguishing good actions from the bad ones. Interestingly, such *inaccurate* $Q$-functions do not prevent the most performant agent to select sub-optimal actions to achieve good performances. Further, we can observe

|  | TD3+BC | CQL | COMBO | IQL |
|---|---|---|---|---|
| halfcheetah-med-v2 | **0.13 (0.01)** | -0.01 (0.01) | 0.01 (0.02) | 0.10 (0.02) |
| halfcheetah-med-rep-v2 | **0.05 (0.01)** | 0.02 (0.01) | 0.02 (0.01) | 0.02 (0.02) |
| halfcheetah-med-exp-v2 | 0.07 (0.04) | 0.08 (0.04) | **0.09 (0.03)** | 0.03 (0.02) |
| hopper-med-v2 | 0.07 (0.03) | 0.02 (0.01) | 0.05 (0.02) | **0.11 (0.03)** |
| hopper-med-rep-v2 | 0.07 (0.05) | 0.08 (0.03) | **0.14 (0.04)** | 0.13 (0.02) |
| hopper-med-exp-v2 | 0.09 (0.07) | 0.11 (0.03) | **0.16 (0.03)** | 0.04 (0.05) |
| walker2d-med-v2 | 0.02 (0.03) | -0.01 (0.01) | -0.01 (0.01) | **0.08 (0.03)** |
| walker2d-med-rep-v2 | 0.10 (0.02) | -0.04 (0.03) | -0.06 (0.05) | **0.10 (0.06)** |
| walker2d-med-exp-v2 | 0.10 (0.04) | 0.09 (0.01) | 0.05 (0.10) | **0.15 (0.04)** |

Table 4: Rank IC measures the ability of $Q$-function to rank different actions

| | TD3+BC | | CQL | | COMBO | | IQL | |
|---|---|---|---|---|---|---|---|---|
| | Top1 Acc | Top3 Acc | Top1 Acc | Top3 Acc | Top1 Acc | Top3 Acc | Top1 Acc | Top3 Acc |
| halfcheetah-med-v2 | **8.72** | **27.60** | 1.81 | 10.40 | 3.84 | 15.61 | 8.11 | 25.91 |
| halfcheetah-med-rep-v2 | **6.70** | **22.11** | 2.86 | 13.53 | 3.67 | 15.43 | 5.92 | 20.20 |
| halfcheetah-med-exp-v2 | 4.60 | 17.17 | 2.62 | 12.19 | 3.86 | 15.28 | 5.84 | 20.59 |
| hopper-med-v2 | 13.20 | 32.08 | 1.52 | 6.92 | 4.67 | 14.48 | **14.90** | **36.21** |
| hopper-med-rep-v2 | 9.70 | 27.39 | 4.51 | 15.05 | 8.33 | 25.63 | **11.41** | **30.76** |
| hopper-med-exp-v2 | 12.43 | 30.57 | 2.77 | 10.41 | 7.11 | 21.19 | **12.98** | **31.62** |
| walker2d-med-v2 | 7.10 | 23.56 | 1.64 | 9.99 | 1.77 | 10.36 | **8.32** | **26.48** |
| walker2d-med-rep-v2 | **8.93** | **28.07** | 2.51 | 11.13 | 3.35 | 13.47 | 8.24 | 26.19 |
| walker2d-med-exp-v2 | 6.50 | 23.38 | 1.10 | 8.72 | 1.93 | 13.47 | **9.70** | **29.86** |

Table 5: TopN accuracy measures the ability of $Q$-function to select the best action.

that the TD3+BC agent and IQL agent usually learn more accurate value functions but achieves worse performance. This suggests that the *policy evaluation* method in TD3+BC and IQL might be more effective but the *policy improvement* method is limited. We provide more discussions in section 6.

> **Observation 2.** A more performant offline RL sometimes learn less accurate $Q$-functions.

## 5 What does a performant policy look like?

In previous sections, we show that a performant offline RL agent sometimes learn relative *low-quality* representations and *inaccurate* value functions. In this section, we try to directly compare the learned policies in order to get a glimpse of "what does a performant offline RL policy look like". To this end, we first design a *policy ranking experiment* to measure *how well* does a performant offline RL policy work in practice. Then we check how often do the SOTA offline RL agents take OOD actions.

### 5.1 Policy ranking experiment

In this experiment, we first use a behavior policy to collect 30K transitions to create a test dataset $\mathcal{D}_{policy}$. At each state $s_i$, we use four baseline agents and a behavior cloning agent to select an action, respectively. Again, we use the online agent to approximate $Q^*(s, a)$ to rank the selected four actions. We use following two metrics to evaluate the policy $\pi_j$ of the $j$-th baseline agent: (1) Average percentage of policy $\pi_j$ that ranks the first/last (with largest/smallest $Q^*$ value) across states. (2) Average MSE of the selected action w.r.t. the optimal action. More details are in Appendix E.1

Experiment results are shown in Table 6 and Table 7. We can observe that the learned policy of COMBO [54] agent is quite *extreme*, which both selects the most optimal and worst actions at the same time. In simple environments such as *halfcheetah* and *hopper*, such behavior helps to attain higher performance. However, in the more complex *walker* environment, the higher percentage of *bad* actions would lead to early termination and thus achieving lower scores. Moreover, as we can observe in Table 7 that the most performant offline RL agent usually have similar or larger MSE loss w.r.t. the optimal action $a^*$. This indicates that a performant policy is good at selecting better actions in different states, even though these sampled actions are still sub-optimal (with high MSE).

> **Observation 3.** A performant offline RL policy needs to strike a balance at selecting good actions while avoiding bad ones. Even though the selected actions are usually still sub-optimal.

|  | TD3+BC | | CQL | | COMBO | | IQL | | BC | |
|---|---|---|---|---|---|---|---|---|---|---|
|  | $P_o$ | $P_w$ | $P_o$ | $P_w$ | $P_o$ | $P_w$ | $P_o$ | $P_w$ | $P_o$ | $P_w$ |
| halfcheetah-med-v2 | 21.05 | 13.02 | 11.18 | 16.33 | **41.10** | **32.86** | 13.82 | 11.96 | 12.85 | 25.82 |
| halfcheetah-med-rep-v2 | 21.89 | 16.72 | 14.45 | 17.40 | **28.10** | **27.09** | 19.42 | 20.16 | 16.15 | 18.63 |
| halfcheetah-med-exp-v2 | 20.59 | 23.55 | 19.65 | 16.07 | **26.57** | **27.50** | 19.17 | 14.89 | 14.02 | 17.98 |
| hopper-med-v2 | 16.84 | 19.98 | 17.23 | 17.63 | **34.10** | **31.29** | 21.20 | 16.35 | 10.63 | 14.74 |
| hopper-med-rep-v2 | 12.76 | 19.52 | 23.25 | 18.68 | **35.01** | **32.28** | 16.13 | 16.40 | 12.86 | 13.12 |
| hopper-med-exp-v2 | 15.23 | 20.22 | 15.98 | 20.55 | **38.46** | **24.74** | 15.44 | 21.18 | 14.89 | 13.32 |
| walker2d-med-v2 | 21.79 | 19.81 | 19.68 | 19.92 | **23.60** | **20.98** | 22.61 | 18.08 | 12.32 | 21.21 |
| walker2d-med-rep-v2 | 20.99 | 16.49 | 15.95 | 20.80 | **26.77** | **32.55** | 23.05 | 15.24 | 13.24 | 14.92 |
| walker2d-med-exp-v2 | 25.58 | 15.64 | 12.66 | 20.70 | **26.37** | **31.41** | 23.74 | 13.09 | 11.65 | 19.16 |

Table 6: Percentage (%) of each agent that selects optimal action ($P_o$) and worst action ($P_w$).

|  | TD3+BC | CQL | COMBO | IQL | BC |
|---|---|---|---|---|---|
| halfcheetah-med-v2 | 8.31 (0.05) | 8.41 (0.04) | **8.27 (0.06)** | 8.30 (0.03) | 8.45 (0.04) |
| halfcheetah-med-rep-v2 | 8.36 (0.03) | 8.57 (0.03) | 8.64 (0.08) | 8.40 (0.04) | **8.35 (0.04)** |
| halfcheetah-med-exp-v2 | 6.96 (0.07) | 7.03 (0.08) | 7.16 (0.16) | **6.85 (0.07)** | 7.11 (0.05) |
| hopper-med-v2 | 2.66 (0.01) | 2.75 (0.04) | 2.77 (0.02) | **2.61 (0.01)** | 2.71 (0.01) |
| hopper-med-rep-v2 | 2.45 (0.10) | 2.65 (0.10) | 2.67 (0.05) | **2.38 (0.09)** | 2.56 (0.08) |
| hopper-med-exp-v2 | 2.22 (0.04) | 2.29 (0.06) | 2.20 (0.07) | 2.20 (0.03) | **2.20 (0.04)** |
| walker2d-med-v2 | 5.39 (0.02) | 5.55 (0.03) | 5.50 (0.02) | **5.38 (0.02)** | 5.48 (0.02) |
| walker2d-med-rep-v2 | **6.10 (0.07)** | 6.61 (0.06) | 6.66 (0.14) | 6.14 (0.05) | 6.34 (0.06) |
| walker2d-med-exp-v2 | 4.65 (0.07) | 4.83 (0.08) | 5.32 (1.06) | **4.61 (0.07)** | 4.73 (0.06) |

Table 7: MSE loss w.r.t. the optimal action.

## 5.2 OOD action experiment

Since existing offline RL algorithms usually attempt to minimize the negative effects caused by OOD samples. Therefore, we try to measure how often would the baseline agents take OOD actions. In the experiment, we utilize a learned dynamics model to predict whether a state-action pair $(s, a)$ is OOD or not. The learned dynamics models is a probabilistic ensemble [8] as in COMBO [54], where each model $T_i(s'|s, a) = \mathcal{N}(\mu_{\theta_i}(s, a), \Sigma_{\theta_i}(s, a))$ outputs a Gaussian distribution with diagonal covariance parameterized by $\theta_i$. We train the probabilistic ensemble using the D4RL dataset. We use the following metric to estimate how likely a state-action pair $(s, a)$ is OOD: the uncertainty [55] estimated by the probabilistic ensemble $\sigma(s, a) = \max_{i=1,\cdots,N} \|\mu_\theta^i(s, a) - \frac{1}{N}\sum_{j=1}^N \mu_\theta^j(s, a)\|_2$.

In particular, we first compute $\sigma(s, a)$ on the original D4RL offline dataset, and then compute the $\sigma(s, a)$ on a dataset collected by the baseline agent. More details are described in Appendix E.2. We report the median of the estimated uncertainty in Table 8. As we can observe that the three model-free offline RL baselines usually learn conservative policies which mostly take high confidence actions with low $\sigma(s, a)$. On the other hand, the model-based baseline COMBO learns policies that sometimes take more risky actions with higher $\sigma(s, a)$. This confirms that taking risky actions in offline RL is a double-edged sword, which sometimes helps to learn from the OOD samples to avoid being too conservative and sometimes incurs the extrapolation error.

**Observation 4.** Taking OOD actions in offline RL could be a double-edged sword that sometimes helps to avoid being over-conservative but sometimes incurs the extrapolation error.

|  | Offline Data | TD3+BC | CQL | COMBO | IQL |
|---|---|---|---|---|---|
| halfcheetah-med-v2 | 10.09 | 7.96 (0.06) | 8.11 (0.15) | **10.25 (0.71)** | 7.96 (0.04) |
| halfcheetah-med-rep-v2 | **23.96** | 15.78 (0.66) | 17.51 (0.88) | 21.54 (1.18) | 16.82 (0.34) |
| halfcheetah-med-exp-v2 | 11.03 | 10.74 (0.86) | 9.17 (1.54) | **23.30 (12.62)** | 8.50 (0.16) |
| hopper-med-v2 | **2.19** | 2.01 (0.02) | 2.00 (0.01) | 1.85 (0.08) | 2.07 (0.03) |
| hopper-med-rep-v2 | **5.13** | 4.48 (1.27) | 3.76 (0.34) | 3.90 (0.84) | 3.59 (0.28) |
| hopper-med-exp-v2 | 1.46 | 1.29 (0.05) | 1.27 (0.02) | **1.52 (0.11)** | 1.32 (0.08) |
| walker2d-med-v2 | **14.01** | 9.14 (0.22) | 8.08 (0.19) | 8.81 (0.99) | 10.73 (0.49) |
| walker2d-med-rep-v2 | **27.59** | 14.21 (1.35) | 13.03 (1.55) | 14.80 (2.04) | 16.09 (0.96) |
| walker2d-med-exp-v2 | 12.87 | 8.86 (0.38) | 8.17 (0.06) | **75.93 (149.39)** | 9.74 (1.37) |

Table 8: The median of the estimated uncertainty $\sigma(s, a)$ in different tasks.

| Baseline | Critic loss function | Actor loss functions |
|---|---|---|
| TD3+BC [16] | $L_{TD}(\mathcal{D}, \theta)$ | $\lambda L_{TD3}(\mathcal{D}, \phi) + \mathcal{C}_{BC}(\mathcal{D}, \phi)$ |
| CQL [36] | $L_{TD}(\mathcal{D}, \theta) + \alpha \mathcal{C}_{CQL}(\mathcal{D}, \theta)$ | $L_{SAC}(\mathcal{D}, \theta, \phi)$ |
| COMBO [54] | $L_{TD}(\mathcal{D}_{\widehat{\mathcal{M}}}, \theta) + \alpha \mathcal{C}_{CQL}(\mathcal{D}_{\widehat{\mathcal{M}}}, \theta)$ | $L_{SAC}(\mathcal{D}_{\widehat{\mathcal{M}}}, \phi)$ |
| IQL [30] | $L_{ER}(\mathcal{D}, \theta)$ | $L_{AWR}(\mathcal{D}, \phi)$ |

Table 9: Different policy evaluation/improvement objectives.

## 6 Case study: relaxed in-sample Q-learning (RIQL)

In this section, we conduct a case study to show that the proposed experiments can be used to compare the effectiveness of existing *policy evaluation/improvement* methods. We first briefly recap the four baseline algorithms as summarized in Table 9. Here we consider a parameterized $Q$-function $Q_\theta^\pi(s, a)$, policy $\pi_\phi(a|s)$ and loss weight parameters $\alpha, \lambda$ that we abuse in different baselines.

**Policy evaluation**. TD3+BC [16] uses the default fitted $Q$-evaluation as in Eq 1. The only difference is that the $\max$ operator over next action $a'$ is replaced by the policy $\pi_\phi(s')$:

$$L_{TD}(\mathcal{D}, \theta) = \mathbb{E}_{(s,a,r,s')\sim\mathcal{D}}\big[(r + \gamma Q_{\hat\theta}^\pi(s', \pi_\phi(s')) - Q_\theta^\pi(s, a))^2\big] \tag{2}$$

CQL [36] further adds a *conservative penalty* to push down large $Q$-values for OOD actions:

$$\mathcal{C}_{CQL}(\mathcal{D}, \theta) = \mathbb{E}_{s\sim\mathcal{D}}\big[\log \sum_a \exp(Q_\theta^\pi(s, a))\big] - \mathbb{E}_{(s,a)\sim\mathcal{D}}\big[Q_\theta^\pi(s, a)\big] \tag{3}$$

COMBO [54] extends CQL by using an augmented dataset $\mathcal{D}_{\widehat{\mathcal{M}}} = \mathcal{D} \cup \mathcal{D}_{model}$ for policy evaluation, where $\mathcal{D}_{model}$ is generated using the learned dynamics model. Unlike other three baselines, IQL [30] uses *expectile regression* (ER) for policy evaluation purely from in-sample data:

$$L_{ER}(\mathcal{D}, \theta) = \mathbb{E}_{(s,a,s',a')\sim\mathcal{D}}\big[L_2^\tau(r(s, a) + \gamma Q_{\hat\theta}^\pi(s', a') - Q_\theta^\pi(s, a))\big] \tag{4}$$

where $L_2^\tau(u) = |\tau - \mathbb{1}(u < 0)|u^2$ and $\tau \in (0, 1)$ is a hyperparameter.

**Policy improvement**. TD3+BC adds a simple behavior cloning loss $\mathcal{C}_{BC}(\phi) = \mathbb{E}[(\pi(s) - a)^2]$ to the original TD3 [17] actor loss $L_{TD3}(\phi) = -\mathbb{E}_{s\sim\mathcal{D}}[Q_\theta^\pi(s, \pi_\phi(s))]$. CQL and COMBO both use the default SAC [22] actor loss function $L_{SAC}(\phi) = \mathbb{E}_{(s,a)\sim\mathcal{D}}[\lambda \log(\pi_\phi(a|s)) - Q_\theta^\pi(s, \pi_\phi(a|s))]$. On the other hand, IQL uses the advantage-weighted regression (AWR) [43] to imitate high-quality in-sample actions $L_{AWR}(\phi) = \mathbb{E}_{(s,a)\sim\mathcal{D}}\big[\lambda \exp((Q_{\hat\theta}(s, a) - V_\psi(s))) \log \pi_\phi(a|s)\big]$, where $V_\psi(s)$ is a value function that only depends on state $s$.

Given these different policy evaluation/improvement methods, there is a lack of clear comparison to show which one is more effective in practice. Here, we try to show that the previously introduced experiment setups can be helpful tools to answer such questions. Recall the results in the Table 4 and Table 5, we can observe that IQL [30] usually achieves higher rank IC and larger topN accuracy, which suggests that the *expectile regression* based *policy evaluation* method in IQL is more capable to learn *accurate* value functions. However, IQL only achieves the best performance in one benchmark task. In addition, Table 8 indicates that IQL learns conservative policy that usually takes high confidence in-sample data. These results suggest that the AWR-based *policy-improvement* method is effective to avoid taking OOD actions but sometimes it is over-conservative which limits the performance. To validate this assumption, we introduce a simple variant of IQL, called *Relaxed In-sample Q-Learning* (RIQL), that replaces the original AWR-based actor loss with:

$$L_{RIQL}(\phi) = L_{SAC}(\phi) - \beta \mathbb{D}_{KL}(\pi_\beta, \pi_\phi) = L_{SAC}(\phi) - \beta \mathbb{E}_{(s,a)\sim\mathcal{D}}[\log \pi_\phi(a|s)]. \tag{5}$$

In short, we add a KL-divergence constraint to the original SAC actor loss, and we drop the $\mathbb{E}_{(s,a)\sim\mathcal{D}}[\log \pi_\beta(a|s)]$ term which is independent of actor parameter $\phi$. The motivation is two-fold: (1) Unlike CQL and COMBO, the learned $Q$-functions in IQL are less discriminating w.r.t. OOD samples. Thus, we need extra policy constraints to avoid taking too many OOD actions. (2) We want to use a less conservative actor loss to enable the policy learning from OOD actions. In addition, Eq 5 is similar to the actor loss in [11]. The difference is that we keep the entropy term which helps to prevent the learned policy from collapsing to a single point. Experiment results of RIQL is shown in Table 10 and more details are described in Appendix F. We can observe that RIQL outperforms IQL in all benchmark tasks, and it achieves higher total evaluation scores than other baseline agents. This result also validates our previous assumption that AWR-based policy improvement method is the bottleneck for IQL in some tasks which makes it over-conservative.

| | TD3+BC | CQL | COMBO | IQL | RIQL |
|---|---|---|---|---|---|
| halfcheetah-med-v2 | 48.89 (0.15) | 46.85 (0.22) | 54.61 (1.48) | 47.43 (0.07) | **55.93 (0.27)** |
| hopper-med-v2 | 60.19 (1.99) | 61.18 (1.16) | **92.20 (5.04)** | 64.64 (3.27) | 91.58 (4.23) |
| walker2d-med-v2 | **84.37 (0.55)** | 81.21 (0.40) | 81.93 (0.66) | 80.28 (2.00) | 80.85 (0.96) |
| halfcheetah-med-rep-v2 | 45.20 (0.30) | 44.95 (0.45) | 52.18 (0.43) | 44.04 (0.71) | **52.39 (0.45)** |
| hopper-med-rep-v2 | 64.96 (7.26) | 88.30 (4.21) | **96.53 (1.91)** | 91.94 (14.94) | 93.13 (7.46) |
| walker2d-med-rep-v2 | 77.24 (1.31) | 77.68 (1.70) | 62.05 (11.46) | 73.53 (7.28) | **81.46 (4.72)** |
| halfcheetah-med-exp-v2 | 89.29 (5.07) | 90.89 (0.44) | 61.53 (9.76) | 88.92 (1.34) | **92.91 (1.18)** |
| hopper-med-exp-v2 | 96.53 (4.97) | **105.19 (2.96)** | 77.57 (16.57) | 91.57 (27.03) | 102.47 (5.49) |
| walker2d-med-exp-v2 | **110.16 (0.18)** | 104.61 (6.39) | 84.98 (42.58) | 107.68 (5.28) | 108.34 (0.58) |
| Total | 676.82 | 700.85 | 663.58 | 690.04 | **759.05** |

Table 10: RIQL achieves better performance than IQL in all benchmark tasks.

# 7 Uncertainty-based sample selection for model-based offline RL

We also try to investigate when does a learned dynamics model is helpful to a model-free offline RL agent? In particular, we reuse the learned dynamics model in Section 5.2 to generate fake samples to train a model-free agent. For example, we introduce an MBRL variant of IQL/TD3+BC, named MIQL/MTD3+BC, which learns from the augmented dataset $\mathcal{D}_{\widehat{\mathcal{M}}} = \mathcal{D} \cup \mathcal{D}_{model}$ as in MOPO [55]. We find that such a naive MBRL agent (a combination of MOPO and model-free offline RL agents) usually performs worse than its model-free counterpart (Table 11) even though they are the same algorithm except for the training data. This result indicates that it is nontrivial to combine a learned dynamics model with model-free offline RL agents due to the model noises. COMBO [54] addresses this problem by adding a conservative penalty on the model-generated samples. Here, we introduce an *Uncertainty-based Sample Selection* (USS) method to filter fake samples with large model noises.

Since it is hard to measure model noises accurately, we adopt the model uncertainty $\sigma(s, a)$ defined in Section 5.2 as a proxy to approximate model noise. In addition, we maintain a dynamic *uncertainty threshold* $\delta_\sigma$ as follows $\delta_\sigma \leftarrow \eta \sigma^q_{batch}(s, a) + (1 - \eta)\delta_\sigma$, where $\sigma^q_{batch}(s, a)$ is the $q$-th quantile of the uncertainties in the sampled fake trajectories. We only add model-generated samples with an uncertainty lower than $\delta_\sigma$ to the model buffer which is later used to train the agent. More details are described in Appendix G. From Table 11, we can observe that the proposed USS trick usually helps to improve the performance of the MIQL/MTD3+BC agent. Moreover, USS is inferior to IQL/TD3+BC in some tasks which shows that it is still challenging to completely solve the model noise problem.

| | IQL | MIQL | MIQL-USS | TD3+BC | MTD3+BC | MTD3+BC-USS |
|---|---|---|---|---|---|---|
| halfcheetah-med-v2 | 47.43 (0.07) | 27.52 (7.69) | **53.85 (0.41)** | 48.89 (0.15) | 1.18 (1.61) | 52.41 (0.39) |
| hopper-med-v2 | 64.64 (3.27) | 8.89 (8.32) | **80.79 (15.21)** | 60.19 (1.99) | 46.07 (7.05) | 67.74 (4.95) |
| walker2d-med-v2 | 80.28 (2.00) | 43.44 (9.59) | 77.21 (4.21) | 84.37 (0.55) | **85.33 (0.95)** | 84.80 (1.09) |
| halfcheetah-med-rep-v2 | 44.04 (0.71) | 44.70 (3.20) | **48.49 (0.30)** | 45.20 (0.30) | 47.80 (0.70) | 47.05 (0.41) |
| hopper-med-rep-v2 | **91.94 (14.94)** | 19.36 (3.18) | 87.63 (6.90) | 64.96 (7.26) | 37.78 (8.71) | 69.64 (8.93) |
| walker2d-med-rep-v2 | 73.53 (7.28) | 59.41 (30.75) | 85.83 (7.59) | 77.24 (1.31) | 85.02 (2.26) | **87.84 (2.57)** |
| halfcheetah-med-exp-v2 | 88.92 (1.34) | 32.47 (4.81) | 82.08 (4.54) | **89.29 (5.07)** | 0.79 (1.36) | 85.63 (2.77) |
| hopper-med-exp-v2 | 91.57 (27.03) | 10.41 (8.66) | 75.88 (41.26) | 96.53 (4.97) | 74.65 (20.27) | **96.81 (4.12)** |
| walker2d-med-exp-v2 | 107.68 (5.28) | 87.96 (9.56) | 108.25 (3.27) | 110.16 (0.18) | 110.37 (0.29) | **110.38 (0.41)** |
| Total | 690.04 | 334.16 | 699.98 | 676.82 | 488.99 | **702.29** |

Table 11: USS is able to effectively mitigate the problem of model noises.

# 8 Conclusion

Besides the rapid development of novel offline RL algorithms, the behaviors of these offline RL agents have not been well studied. In this work, we take a closer look at SOTA offline RL agents. Specifically, we introduce a series of experiments to compare the learned representations, value functions and policies of four baseline agents. Surprisingly, we find that a performant agent sometimes has relatively low quality representations and inaccurate value functions. We also show that the proposed experiment setups can be used to compare the effectiveness of different policy evaluation/improvement methods by introducing a relaxed version of IQL that achieves SOTA performance. Lastly, we investigate when a learned dynamics model can help model-free offline RL agents, and we introduce an uncertainty-based sample selection method to mitigate the problem of model noises.

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
