# Appendix

In the appendix, we first provide more background information and introduce some related works. Then we report the details of experiment setups and additional experiment results on all test environments. Lastly, we discuss the limitations and future directions of this work.

## A    Extended background material

Here, we provide a more extensive background review. Specifically, we present more discussions on offline reinforcement learning and representation learning for reinforcement learning.

### A.1    Offline reinforcement learning

Offline reinforcement learning (RL) aims to learn effective control policies purely from fixed pre-collected datasets. Recent years have seen a surge of different offline RL methods. We crudely classify some latest works in the following categories.

**Policy-regularized offline RL.** The majority of the recent proposed offline RL algorithms share a similar idea – constraining the learned policy to stay close to the behavior policy. For example, BCQ [18] used a variational auto-encoder (VAE) to generate actions that are similar to samples in the offline dataset. KL-Control [24] and CDC [11] used KL divergence, BEAR [35] adopted maximum mean discrepancy (MMD) divergence, FBRC selected Fisher-divergence [31] as a regularizer in the loss function. On the other hand, BRAC[50] and BRAC+ [56] proposed general frameworks for the such policy-regularized approaches.

**Pessimism-based offline RL.** Another line of research focuses on attacking the over-estimation problem directly by using conservative penalties. CQL [36] proposed a penalty regularizer for out-of-distribution (OOD) samples, such that we can learn conservative Q-functions that lower-bounds its true value. In addition, Buckman et al. [5] proposed a general theoretical framework to unify some existing pessimism-based approaches. Recently, CDC [11] introduced a new regularizer that only penalizes high-valued samples selected by the actor.

**Generalized behavior cloning (BC).** Some other works resort to weighted behavior cloning, which reduces the RL problem to a supervised learning problem [10]. For example, AWR [43] and CRR [49] used value-weighted regression to filter high-quality samples. BAIL [7] proposed upper-envelope to select good actions for later imitation learning. Recently, IQL [30] performed weighted behavioral cloning for policy extraction.

**Model-based offline RL.** One motivation for using Model-based RL (MBRL) in the offline setting is to increase the data coverage, where the batch RL agent can learn policies with OOD states [55, 29]. MOPO [55] and MoREL [29] measured the uncertainty of the model's prediction to formulate an uncertainty-penalized MDP. COMBO [54] combined CQL with a learned model by penalizing the samples generated by the learned model. MuZero Unplugged [47] directly used the learned model for policy and value improvement through planning.

**Other types of offline RL.** In addition to using uncertainty estimation under the MBRL framework, some recent model-free batch RL algorithms also adopt the uncertainty to mitigate the over-estimation error. UWAC [52] used dropout-uncertainty estimation to down-weight OOD samples loss. MSG [19] used an ensemble of independently-updated Q-functions for uncertainty estimation. Furthermore, REM [1] showed that standard off-policy RL methods can achieve good performance when the dataset is large and diverse.

### A.2    Representation learning for reinforcement learning

Some recent works introduced different representation metrics as proxies to evaluate the quality of the learned representations of an RL agent [34, 33, 38, 41]. We provide a more detailed introduction to the following three metrics that we use in the experiments.

**Definition 1** (Feature dot-product). *Feature dot-product $\phi(s,a)^\top \phi(s',a')$ is the dot-product of two critic representations, where $s' \sim \mathcal{P}(\cdot|s,a)$ and $a' \sim \pi(\cdot|s')$ is the next state and next action.*

**Definition 2** (Effective rank). *Effective rank* $\mathrm{srank}_\delta(\Phi) = \min\left\{ k : \frac{\sum_{i=1}^{k} \sigma_i(\Phi)}{\sum_{i=1}^{d} \sigma_i(\Phi)} \geq 1 - \delta \right\}$ *of a feature matrix* $\Phi \in \mathbb{R}^{|\mathcal{S}| \cdot |\mathcal{A}| \times d}$ *approximates the rank of* $\Phi$ *[33], where* $\{\sigma_i(\Phi)\}$ *are the singular values of* $\Phi$ *in the decreasing order* ($\sigma_1 \geq \cdots \geq \sigma_d \geq 0$) *and* $\delta$ *is a threshold parameter, i.e.,* $0.01$.

**Definition 3** (Effective dimension). *Effective dimension* $d_{\mathrm{eff}}(\Phi) = N \max_{i=1,\cdots,N} \|P_\Phi e_i\|_2^2$ *of a feature matrix* $\Phi \in \mathbb{R}^{|\mathcal{S}| \cdot |\mathcal{A}| \times d}$ *measures the sparsity of the column space of* $\Phi$ *[38], where* $N = |\mathcal{S}| \cdot |\mathcal{A}|$ *and* $P_\Phi$ *is the orthogonal projector onto the column space of* $\Phi$.

DR3 [34] introduced the *feature co-adaptation* phenomenon where the *feature dot-product* keeps increasing during the training of a deep $Q$-Network. In particular, DR3 provided a theoretical analysis to show that optimizing TD errors with SGD has an implicit "regularizer" that leads to the feature co-adaptation phenomenon which usually hurts the performance. Later, some related works [33, 38, 41] introduced the *representation collapse* phenomenon, where the rank of the feature space shrinks during the training step, which limits the learning capacity of an RL agent. In particular, Kumar et al. [33] pointed out that such representation collapse phenomenon is connected to the bootstrapping based update, and provided some theoretical explanations in the context of kernel regression and deep linear networks. On the other hand, Lan et al. [38] provided an informative bound on the generalization error w.r.t the *effective dimension* of state representations, and showed that a smaller *effective dimension* helps to improve the performance. To mitigate the problem of representation collapse, InFeR [41] presented a regularization term that forces the trained networks to regress to a random function of their initial output.

Another closely related work is ACL [53], which empirically investigated a number of representation auxiliary losses for RL agents. Moreover, Deng et al. [9] also explored the representation bottlenecks for deep neural networks. A major difference of this work and these related works is that we studied the detailed behaviors, including representations, value function and policies, of offline RL agents.

## B  Experiment details

In this section, we present more details about the dataset and the baseline algorithms.

### B.1  Experiment setups

In the experiment, we evaluate all baseline agents on the standard D4RL benchmark [14]. In specific, we use the "-v2" dataset, which contains more metadata and fixes bugs in the "-v0" dataset [1], to train each baseline agent for 1M steps. In all experiments, we report the average result and standard deviation over 5 random seeds. We run all experiments on a workstation with GeForce GTX 3090 GPU and an Intel Core i9-12900KF CPU.

### B.2  Baseline algorithms

We re-implement each baseline algorithm in JAX [13]. For TD3BC [16], CQL [36] and IQL [30], we use the default parameter settings as in the original implementations. Moreover, we find the default parameters in COMBO [54], which is originally trained on the "-v0" dataset, sometimes fail in the "-v2" dataset. Thus, we additionally tune the conservative parameter $\beta$ for COMBO using the offline cross-validation scheme as described in the original paper [54].

For the software, we use the following versions:

- Python 3.9
- Jax 0.3.10
- Gym 0.23.1
- Mujoco 2.1.2
- mujoco-py 2.1.2.14
- d4rl 1.1

---

[1] https://github.com/rail-berkeley/d4rl/wiki/Tasks

The running time for 1M steps for each baseline algorithm in the JAX version is usually faster than the original implementations [30] as summarized in Table 12:

| | TD3BC | CQL | COMBO | IQL |
|---|---|---|---|---|
| Running time (min) | 15 | 90 | 70 | 20 |

Table 12: Running time for 1M steps.

## C    Representation evaluation experiments

In this section, we provide more results of the representation experiments in different environments. In the experiment, we first train an online TD3 [17] agent for 2M steps to approximate the optimal policy $\pi^*(a|s)$ and optimal value functions $Q^*(s, a)$ and $V^*(s)$. For each probing target, we train a linear model for 200 epochs using the Adam optimizer with learning rate 3e-4. We adopt an early stopping strategy with patience of 10 epochs. We report the average result of a 5-fold cross-validation. The most performant baseline agents in each environment are brown, and the agent with the best probing experiment result is in **bold**.

### C.1    Representation probing experiment results

In the experiment, the COMBO agent crashed in the *walker2d-medium-expert-v2* environment for one random seed (learned explosive $Q(s, a)$). In some probing experiments, the crashed COMBO agent diverges. We report the mean and standard deviation of the MSE loss over 5 random seeds.

From Table 13, Table 14 and Table 15, we can observe that the TD3BC agent usually has good probing experiment results (with the least probing loss), however, the TD3BC agent only achieves the best performance in 2 of the 9 environments. These results show that the transition dynamics-based side information is not that important for offline RL agents in the selected D4RL tasks. Intuitively, the most performant agent usually has small optimal $a^*$ probing loss (Table 16). This indicates that as long as the actor representation $\psi(s)$ preserves the ability to learn good actions (low optimal action probing loss), then the offline RL agent holds the potential to achieve good performance. From Table 17 and Table 18, we can find that the probing loss w.r.t. the optimal value functions are usually very large. Such large probing loss highlights the difficulty to learn accurate value functions in the offline setting due to the limited data coverage and additional policy/value constraints.

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

Table 17: Optimal $Q^*(s, a)$ probing experiment: each number is divided by 1000.

| | TD3BC | CQL | COMBO | IQL |
|---|---|---|---|---|
| halfcheetah-med-v2 | 76.30 (22.05) | 123.11 (42.42) | **48.73 (35.06)** | 100.97 (43.36) |
| halfcheetah-med-rep-v2 | 109.73 (13.14) | 95.85 (7.56) | **76.23 (9.11)** | 153.07 (11.04) |
| halfcheetah-med-exp-v2 | 41.72 (13.59) | 79.86 (26.85) | **23.90 (13.68)** | 186.22 (20.99) |
| hopper-med-v2 | 2.76 (0.52) | 5.82 (3.92) | **1.10 (0.29)** | 3.38 (1.22) |
| hopper-med-rep-v2 | **1.73 (0.50)** | 1.93 (0.83) | 2.13 (0.43) | 2.30 (0.90) |
| hopper-med-exp-v2 | 1.78 (0.85) | 4.79 (2.32) | **1.15 (0.13)** | 1.90 (0.53) |
| walker2d-med-v2 | 3.36 (0.50) | 3.33 (0.39) | **2.99 (0.44)** | 4.35 (0.37) |
| walker2d-med-rep-v2 | 3.64 (0.12) | 4.49 (0.71) | **2.27 (0.34)** | 5.67 (0.25) |
| walker2d-med-exp-v2 | **3.98 (0.56)** | 4.09 (0.31) | 4.80 (2.97) | 4.27 (0.32) |

Table 18: Optimal $V^*(s)$ probing experiment: each number is divided by 1000.

## C.2 Representation metric experiment results

Before reporting the results for the representation metric experiment, we first give a brief discussion of the implementation of the *effective dimension*. As shown in the following code snippet (Algo 1), the original implementation of the *effective dimension* [2] uses a hard threshold (1e-5) to approximate the rank of the feature matrix. However, such a hard threshold is prone to over-estimate the rank when the norm of the representation is large, as we will see in Table 19 and Table 20. Therefore, we use a modified implementation which approximate the rank using a relative threshold (effective rank).

---

**Algorithm 1** A comparison of two implementations of effective dimension.

```
# matrix: sample representations (Nxd)
# thresh: a threshold to approximate the matrix rank

def calculate_effective_dim1(matrix, thresh=1e-5):
    """The original implementation"""
    num_rows, _ = matrix.shape

    # SVD
    u, s, _ = np.linalg.svd(matrix, full_matrices=False, compute_uv=True)

    # approximate matrix rank using a hard threshold
    rank = max(np.sum(s >= thresh), 1)

    # approximate the effective dimension
    u1 = u[:, :rank]
    projected_basis = np.matmul(u1, np.transpose(u1))
    norms = np.linalg.norm(projected_basis, axis=0, ord=2) ** 2
    eff_dim = num_rows * np.max(norms)
    return eff_dim

def calculate_effective_dim2(matrix, thresh=0.99):
    """A modified implementation"""
    num_rows, _ = matrix.shape

    # normalize the matrix
    normalized_matrix = matrix / np.linalg.norm(matrix, axis=-1, keepdims=True)

    # SVD
    u, s, _ = np.linalg.svd(normalized_matrix, full_matrices=False, compute_uv=True)

    # approximate matrix rank using a relative threshold (effective rank)
    cumsum_s = s.cumsum()
    threshold = cumsum_s[-1] * thresh
    rank = sum(cumsum_s <= threshold)

    # approximate the effective dimension
    u1 = u[:, :rank]
    projected_basis = np.matmul(u1, np.transpose(u1))
    norms = np.linalg.norm(projected_basis, axis=0, ord=2) ** 2
    eff_dim = num_rows * np.max(norms)
    return eff_dim
```

---

From Table 19, we can observe that the critic constraint in offline RL will make the *feature co-adaption* more severe. TD3BC agent does not use any critic constraint, and the consecutive state-action pairs

---

[2]Original implementation of the effective dimension.

are the least similar. On the other hand, the *feature co-adaptation* phenomenon is particularly severe when we adds a conservative penalty. Interestingly, such a large feature dot-product does not prevent the COMBO agent to achieve good performance in some offline RL tasks. In addition, combined results in Table 19 and Table 20, we can conclude that the increasing feature dot-product is mainly caused by the representation norm. These results also provide a hint for how the conservative penalty in CQL or COMBO works. As we can see that the representation norm $|\phi(s, a)|$ is usually large for a CQL or COMBO agent, then it learns a lower-bounded $Q$-values $Q_\theta(s, a) = \theta^\top \phi(s, a)$ by a small $|\theta|$ and (or) dissimilar vector direction of $\theta$ and $\phi(s, a)$.

From Table 21 and Table 22, we can observe that the COMBO agent ususally suffers from the *representation collapse* problem while its model-free counterpart CQL agent doesn't. This result indicates that the usage of a learned dynamics model would sometimes affect the learned critic representations. We leave a deeper investigation for future work.

|  | TD3BC | CQL | COMBO | IQL |
|---|---|---|---|---|
| halfcheetah-med-v2 | 10.34 (0.86) | 1036.34 (1179.43) | 5115.07 (3688.67) | **7.06 (0.31)** |
| halfcheetah-med-rep-v2 | **4.55 (0.41)** | 81.10 (9.94) | 229.43 (66.82) | 7.11 (1.10) |
| halfcheetah-med-exp-v2 | 7.98 (0.26) | 469.44 (57.80) | 750.29 (495.01) | **6.67 (0.69)** |
| hopper-med-v2 | 8.43 (1.40) | 56.37 (21.31) | 335.85 (301.72) | **4.93 (0.31)** |
| hopper-med-rep-v2 | **3.60 (0.58)** | 44.93 (9.62) | 2048.59 (1415.81) | 4.82 (0.58) |
| hopper-med-exp-v2 | 6.81 (1.03) | 62.69 (3.59) | 1031.57 (671.65) | **4.86 (0.11)** |
| walker2d-med-v2 | **5.80 (1.26)** | 123.04 (61.53) | 44.07 (11.08) | 8.76 (0.86) |
| walker2d-med-rep-v2 | **3.40 (0.34)** | 93.49 (3.73) | 144.16 (61.59) | 5.90 (0.36) |
| walker2d-med-exp-v2 | **7.01 (0.93)** | 128.92 (12.80) | NaN | 7.31 (0.37) |

Table 19: Dot-product for $\phi(s, a)$: each value is divided by 1000.

|  | TD3BC | CQL | COMBO | IQL |
|---|---|---|---|---|
| halfcheetah-med-v2 | **0.74 (0.03)** | 0.85 (0.08) | 1.00 (0.00) | 0.95 (0.01) |
| halfcheetah-med-rep-v2 | **0.62 (0.03)** | 0.77 (0.02) | 0.94 (0.02) | 0.86 (0.02) |
| halfcheetah-med-exp-v2 | **0.74 (0.01)** | 0.96 (0.01) | 0.96 (0.02) | 0.89 (0.01) |
| hopper-med-v2 | **0.81 (0.03)** | 0.99 (0.01) | 1.00 (0.00) | 1.00 (0.00) |
| hopper-med-rep-v2 | **0.65 (0.05)** | 0.99 (0.00) | 1.00 (0.00) | 1.00 (0.00) |
| hopper-med-exp-v2 | **0.81 (0.04)** | 0.99 (0.00) | 0.99 (0.01) | 1.00 (0.00) |
| walker2d-med-v2 | **0.72 (0.05)** | 0.95 (0.02) | 0.99 (0.00) | 0.97 (0.00) |
| walker2d-med-rep-v2 | **0.65 (0.02)** | 0.97 (0.00) | 1.00 (0.00) | 0.95 (0.00) |
| walker2d-med-exp-v2 | **0.75 (0.03)** | 0.99 (0.00) | 0.99 (0.01) | 0.98 (0.00) |

Table 20: Cosine similarity for $\phi(s, a)$.

|  | TD3BC | CQL | COMBO | IQL |
|---|---|---|---|---|
| halfcheetah-med-v2 | 131.40 (6.58) | **145.00 (9.51)** | 6.80 (1.48) | 144.60 (5.41) |
| halfcheetah-med-rep-v2 | 165.40 (0.89) | **174.00 (7.52)** | 84.80 (10.28) | 164.40 (3.51) |
| halfcheetah-med-exp-v2 | 131.00 (3.61) | 129.20 (6.76) | 73.60 (10.53) | **147.80 (4.15)** |
| hopper-med-v2 | 47.00 (4.00) | **110.40 (9.50)** | 32.80 (11.10) | 56.00 (4.74) |
| hopper-med-rep-v2 | 61.80 (4.44) | **93.60 (3.05)** | 32.20 (6.14) | 71.80 (4.02) |
| hopper-med-exp-v2 | 51.00 (4.74) | **110.40 (6.43)** | 28.00 (9.75) | 61.60 (7.99) |
| walker2d-med-v2 | 121.60 (5.98) | **155.60 (15.60)** | 142.00 (7.18) | 133.60 (1.67) |
| walker2d-med-rep-v2 | 168.00 (3.16) | **169.60 (5.18)** | 49.00 (8.51) | 162.00 (2.65) |
| walker2d-med-exp-v2 | 126.60 (1.67) | **151.60 (2.30)** | 68.20 (37.77) | 138.20 (3.63) |

Table 21: Effective rank for $\phi(s, a)$.

|  | TD3BC | CQL | COMBO | IQL |
|---|---|---|---|---|
| halfcheetah-med-v2 | **1238.21 (206.54)** | 1671.35 (266.73) | 7506.93 (4914.83) | 2853.29 (662.05) |
| halfcheetah-med-rep-v2 | **964.80 (74.90)** | 3041.89 (1472.68) | 6121.73 (2351.42) | 1331.98 (50.85) |
| halfcheetah-med-exp-v2 | **1694.37 (430.47)** | 2340.58 (549.36) | 4009.00 (933.79) | 2199.08 (322.87) |
| hopper-med-v2 | 2994.54 (792.37) | **1991.76 (240.80)** | 3493.77 (857.89) | 3143.34 (524.72) |
| hopper-med-rep-v2 | 2853.40 (349.95) | 2771.78 (619.28) | **1902.39 (347.89)** | 2627.52 (567.50) |
| hopper-med-exp-v2 | 2745.54 (604.30) | **2240.43 (926.86)** | 3906.44 (2240.04) | 2672.30 (586.31) |
| walker2d-med-v2 | **1854.38 (288.18)** | 2460.55 (1292.07) | 2588.08 (844.80) | 1973.46 (396.32) |
| walker2d-med-rep-v2 | 1535.55 (82.77) | 1558.29 (184.18) | 11855.81 (1549.88) | **1478.16 (156.08)** |
| walker2d-med-exp-v2 | 1859.44 (378.47) | 2610.34 (1587.02) | **1741.50 (989.31)** | 1914.29 (142.38) |

Table 22: Effective dimension for $\phi(s, a)$.

# D  Value ranking experiments

Fig 5 illustrates the pipeline of value ranking experiment. Firstly, we use a behavior policy $\pi_\beta$ to interact with the environment to collect a set of states $\{s_1, \cdots, s_N\}$ as the test dataset. In particular, the behavior policy $\pi_\beta$ is a mixture of four baseline offline RL agents and a near-optimal online agent. We use each agent to collect 50K transitions. Secondly, at each state $s_i$, we use each baseline agent to select $m = 5$ different actions $a' = \pi(\cdot|s_i) + \epsilon$, where $\epsilon_i$ is a Gaussian noise with $\mu = 0$ and $\sigma = 0.2$. Thirdly, we use each baseline agent to evaluate the $Q$-values of the collected state-action pairs. Lastly, we use the online agent to approximate the optimal agent to compute the Spearman's rank correlation coefficient (rank IC) and top-N accuracy.

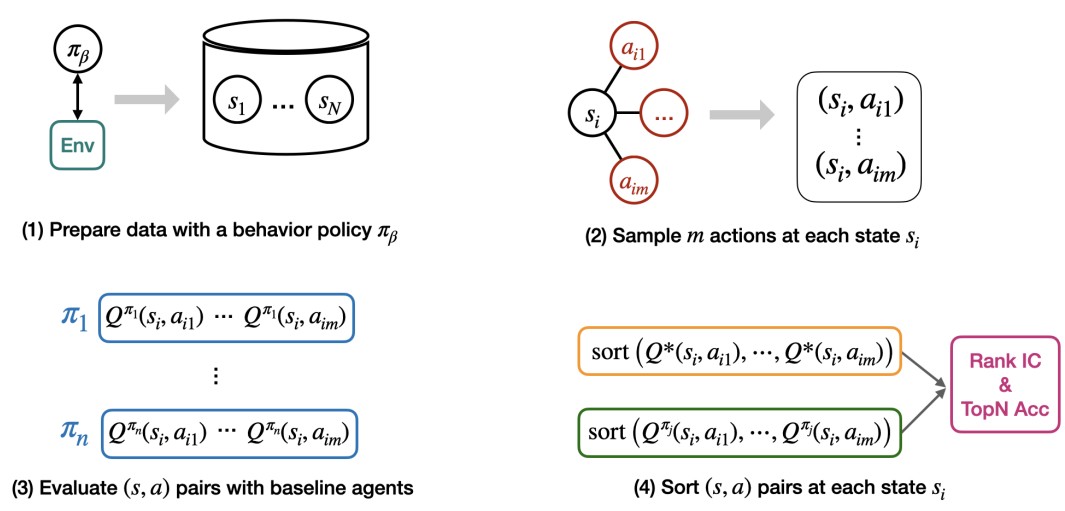

Figure 5: The pipeline of the value ranking experiment.

Since this experiment only utilizes the learned $Q$-function of each baseline agent, therefore, we use the experiment result as a proxy to evaluate how *accurate* the learned $Q$-values are at ranking different actions. Moreover, the experiment result is also a useful evidence to show the effectiveness of different policy evaluation methods.

# E  Performant policy experiments

## E.1  Policy ranking experiment

Fig 6 shows the pipeline of the policy ranking experiment. Similar to the previous value ranking experiment, we use the online agent to approximate optimal $Q^*(s, a)$ to rank the actions selected by each baseline agents. We use the following two metrics to evaluate the *goodness* of the learned

policy: (1) Average percentage of policy $\pi_j$ that rank the first/last (with largest/smallest $Q^*$ value) across all states $\frac{1}{|\mathcal{D}_{policy}|}\sum_i \mathbb{1}\{\mathrm{rank}(Q^*(\pi_j(s_i)), s_i)) = 1\}$. (2) Average mean square error (MSE) loss of the selected action w.r.t. the optimal action $\frac{1}{|\mathcal{D}_{policy}|}\sum_t \|a_i^* - \pi_j(s_i)\|^2$.

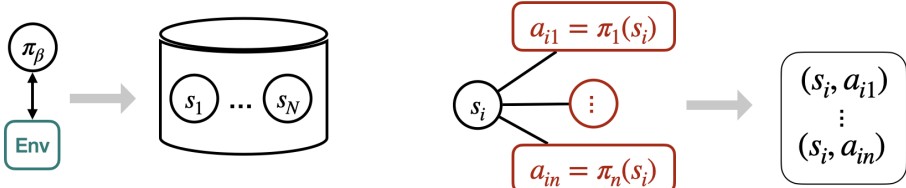

**(1) Prepare data with a behavior policy $\pi_\beta$**     **(2) Sample action using base agents' policies**

$$\mathrm{sort}\left(Q^*(s_1, a_{11}), \cdots, Q^*(s_1, a_{1n})\right)$$
$$\vdots$$
$$\mathrm{sort}\left(Q^*(s_N, a_{N1}), \cdots, Q^*(s_N, a_{Nn})\right)$$

**(3) Ranking policy using the optimal Q\* function**

Figure 6: The pipeline of the policy ranking experiment.

### E.2 OOD action experiments

To investigate how often do different offline RL agents take OOD actions, we first use each baseline agent to interact with the environment to collect 50K transitions. Then we use a probabilistic ensemble-based dynamics model [8] as in COMBO [54] to estimate the uncertainty [55] of each state-action pair as $\sigma(s, a) = \max_{i=1,\cdots,N} \|\mu_\theta^i(s, a) - \frac{1}{N}\sum_{j=1}^N \mu_\theta^j(s, a)\|_2$. In the experiment, we set the ensemble model number $N = 7$ as in COMBO. We report the average median and standard deviation of the uncertainty in the collection samples in Table 8.

## F    Relaxed in-sample Q-Learning (RIQL)

In RIQL, we attempt to make a minimal modification w.r.t. the IQL algorithm [30]. Since the motivation of RIQL is to relax the in-sample constraint for the policy improvement step in IQL, we therefore replace the *Advantage Weighted Regression* (AWR) loss function with the following:

$$L_\pi(\phi) = L_{SAC}(\phi) - \beta\mathbb{E}_{(s,a)\sim\mathcal{D}}[\log \pi_\phi(a|s)]$$

In short, we add a KL-divergence constraint to the SAC [22] actor loss. We determine the parameter $\beta = \{0.1, 0.25, 0.5, 1.0, 1.5\}$ using the offline cross-validation scheme as described in COMBO [54]. For the policy evaluation step, we follow the original expectile regression-based method to update the two value functions $Q_\theta(s, a)$ and $V_\psi(s)$:

$$L_Q(\theta) = \mathbb{E}_{(s,a,s')\sim\mathcal{D}}\big[(r(s, a) + \gamma V_\psi(s') - Q_\theta(s, a))^2\big]$$
$$L_V(\psi) = \mathbb{E}_{(s,a)\sim\mathcal{D}}[L_2^\tau(Q_{\hat\theta}(s, a) - V_\psi(s))]$$

where $Q_{\hat\theta}(s, a)$ is the target $Q$ network, $L_2^\tau(u) = |\tau - \mathbb{1}(u < 0)|u^2$ and $\tau \in (0, 1)$ is a hyperparameter. The pseudocode of RIQL is described in Algo 2. In addition, Fig 7 shows the learning curves of RIQL where we report the average evaluation return over 5 random seeds and the standard deviation is in shadow.

---

**Algorithm 2** Relaxed In-sample Q-Learning (RIQL)

---
**Input:** offline dataset $\mathcal{D}$.

Initialize parameters $\psi, \theta, \hat{\theta}, \phi$.

Policy evaluation with expectile regression:

**for** each gradient step **do**

$\quad \psi \leftarrow \psi - \lambda_V \nabla_\psi L_V(\psi)$

$\quad \theta \leftarrow \theta - \lambda_Q \nabla_\theta L_Q(\theta)$

$\quad \hat{\theta} \leftarrow (1 - \alpha)\hat{\theta} + \alpha\theta$

**end for**

Policy improvement with KL constraint:

**for** each gradient step **do**

$\quad \phi \leftarrow \phi - \lambda_\pi \nabla_\phi L_\pi(\phi)$

**end for**

---

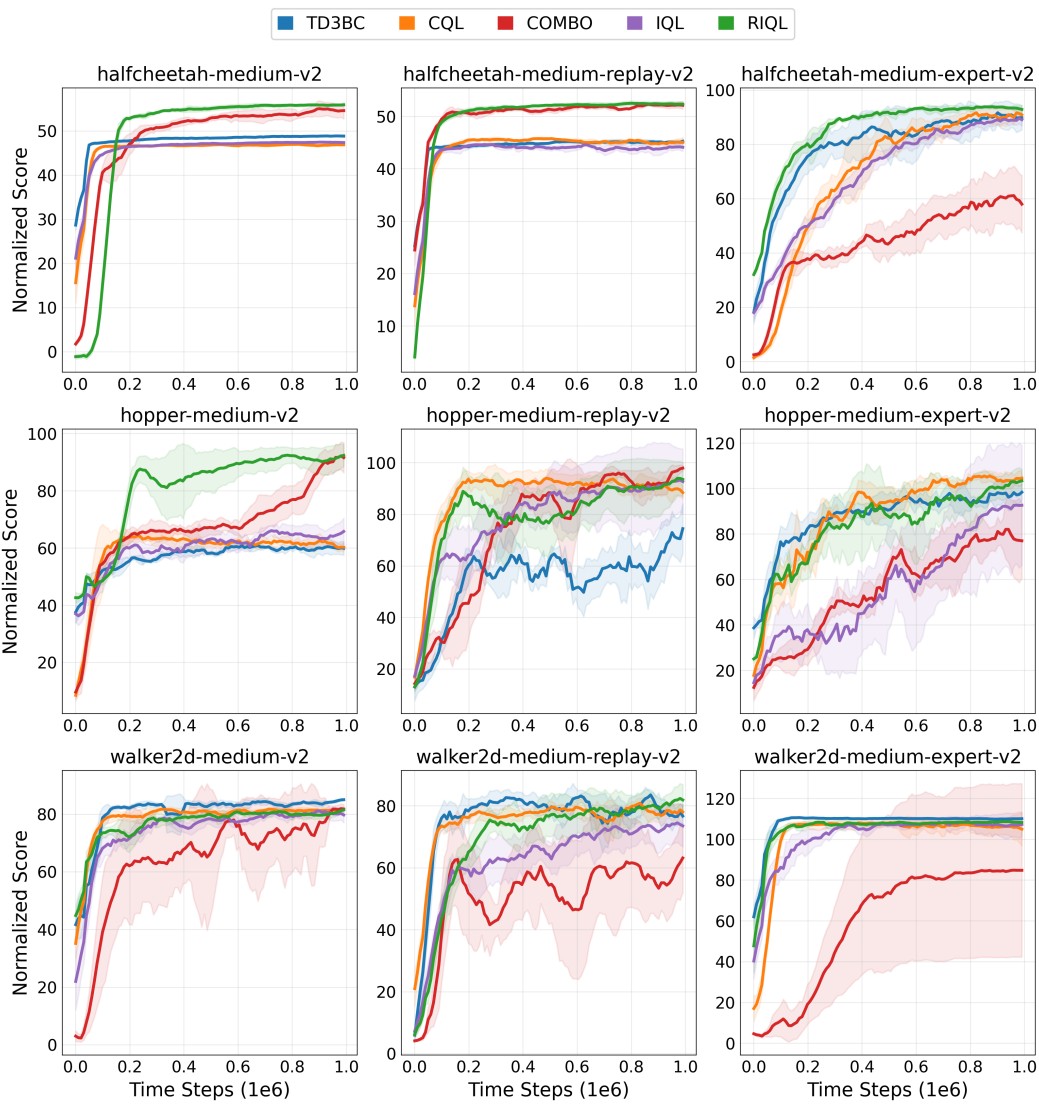

Figure 7: Learning curve of different offline RL agents. We report the average mean and standard deviation over 5 random seeds.

# G   Uncertainty-based Sample Selection (USS)

The proposed *Uncertainty-based Sample Selection* (USS) is a simple trick for model-based offline RL. Here we present a detailed pipeline to showcase how it can be implemented.

- We first learn a probabilistic ensemble-based dynamics model as in the COMBO using the offline dataset.
- For every $T = 1000$ step, we use the offline RL agent to rollout the model to generate 10K short trajectories (with horizon = 5).
- We then use the generated samples to update the dynamic uncertainty threshold $\delta_\sigma \leftarrow \eta\sigma^q_{batch}(s, a) + (1 - \eta)\delta_\sigma$, where $\eta = 0.1$ and $\sigma^q_{batch}(s, a)$ is the median uncertainty of the sampled transitions. The $\delta_\sigma$ is initialized to be the median uncertainty of the transition samples in the first rollout. We only add samples with uncertainty lower than $\delta_\sigma$ to the extra model buffer $\mathcal{D}_{model}$.
- We train the agent using the augmented dataset $\mathcal{D} \cup \mathcal{D}_{model}$ where $\mathcal{D}$ is the original offline dataset.

# H   Limitations and future directions

In this work, we introduced a series of experiment setups to analyze the behaviors of offline reinforcement learning agents. Specifically, we focus on three fundamental aspects of an RL agent – learned representations, value functions, and policies. From the empirical experiment results, we discover some bottlenecks of current SOTA offline RL agents. As a case study, we introduced a variant of IQL agent, called RIQL, which uses a relaxed policy improvement method and achieves better performance. A major limitation of our work is that we need to use an online agent to approximate the optimal agent, which might be unavailable in many problems. In this work, we empirically show that some performant offline RL agents learned low-quality representations. Therefore, an interesting future direction is to investigate the effectiveness of the latest representation learning methods in offline RL. Another future direction is to extend the proposed experiment setups to the online setting.