# OpenReview forum: "A Closer Look at Offline RL Agents"
_NeurIPS.cc/2022/Conference — NeurIPS 2022 Accept_

### Official Review · Reviewer_LmNq · 2022-06-22

**Rating:** 6
**Confidence:** 5
**Soundness:** 2 fair
**Presentation:** 3 good
**Contribution:** 3 good

**Summary:**

Introduces a set of experiments to evaluate & diagnose the bottleneck of offline RL algorithms along three axes:
(1) Representations
- Representation probing: Use the second last layer of the critic network to predict the next state, reward, and action via linear regression. Similarly, use the actor embedding to predict the optimal action, value function, and Q-function (which is approximated by a trained TD3 policy).
- Representation metric: Feature dot-product and effective rank of critic representations.

(2) Value functions
- Rank actions using the learned Q(s, a)

Observation: TD3BC and IQL learn more accurate value functions (policy evaluation might be more effective), but achieve worse performance (policy improvement might be limited). More generally,  an offline RL algorithm can be higher-performing, but learn poor representations and value functions.

(3) Policies
- Policy ranking experiment: Average MSE of the selected action vs. optimal action.
- How often does the policy take OOD actions?

Observations:
- COMBO selects most optimal & worst actions.
- Performant policy is good at selecting better actions, even if the actions are sub-optimal.

The paper also introduces a new offline RL algorithm called RIQL (Relaxed In-Sample Q-learning), based on a simple modification of IQL.
- The method is motivated by empirical, heuristic-based observations: that AWR-based policy improvement is effective to avoid taking OOD actions, but is sometimes over-conservative. It add extra policy constraints and use a less conservative actor loss to enable learning from OOD actions.
- Note that there is no theoretical justification or provable guarantees for the method.

The paper also investigates when a learned dynamics model helps model-free offline RL. Introduces an uncertainty-based sample selection method that is more robust to model noises.


**Questions:**

1. Is it possible to add a more different task that is known to be more challenging than the standard locomotion tasks (halfcheetah, hopper, walker2d)?

2. Description of the policy ranking metric in L89/L637 is unclear: “Average percentage of policy pi_j that ranks the first/last (with largest/smallest Q* value) across states.” Did you instead mean: “Percentage of states where the Q*(pi_j(s), s) is largest/smallest out of the baseline and BC agents”?


--Minor typos that did not affect this review in any way--

L35: “When does a learned dynamics model is helpful to a model-free offline RL agent” -> “When is a learned dynamics model helpful to a model-free offline RL agent?”

L162: “rank the M action” -> “rank the M actions” ?

L228: “IQL uses expectile regression (ER) to for policy evaluation” -> “IQL uses expectile regression (ER) for policy evaluation”

There are many duplicate citations, e.g.:
- [18] and [19]
- [33] and [34]
- [38] and [39]
- [43] and [44]
- [54] and [55]


**Limitations:**

Yes

**Strengths And Weaknesses:**

Strengths:
The paper presents thoughtful experiments analyzing different offline RL algorithms. It is an interesting result that performant offline RL algorithms often exhibit poor representations and inaccurate value functions. (However, I wonder if this is due to the environments/tasks, which are all similar, state-based locomotion tasks.) Overall, I think the empirical experiments & analyses are useful for better understanding offline RL algorithms on simple classic control tasks, although it is unclear if these analyses still hold for more challenging tasks.

Weaknesses:
1. My main concern is that all evaluations are done on the toy classic control locomotion tasks in the D4RL benchmark, which are limited due to the fact that simple filtered behavioral cloning outperforms SOTA offline RL methods on D4RL; and therefore, good performance on these simple D4RL tasks does not necessarily translate to good offline RL performance. The paper's contribution would be greatly improved if the authors add a more challenging and different task other than toy locomotion, such as D4RL AntMaze navigation, manipulation tasks, or image-based tasks such as Atari.

2. The proposed method (RIQL) performs similarly as other baseline methods on the toy locomotion tasks (Table 10), which shows that RIQL is a reasonable offline RL algorithm to use. On the other hand, RIQL is a heuristic-based modification of existing offline RL objectives, and there is no theoretical justification (e.g. provable guarantees about policy improvement) for the proposed method RIQL.

---

> ### Author Response · Authors · 2022-08-02
> **Response to Reviewer LmNq**
>
> Thank you for the careful read of the paper, and the positive assessment of the significance of this work. We respond to specific questions and comments below.
>
> > **Q1:** experiments are only done on classic control locomotion tasks
>
> Thanks for the suggestions. We added experiments on random datasets (Q2 Reviewer HrJL) and antmaze tasks (Q1 Reviewer LgJQ). Here, we also tried on the image-based control tasks. Due to the computation complexity, we only conduct two case studies on MinAtar and Atari games.  We would be glad to add results on more environments in the later. In particular, we compare following baseline algorithms:
>
> - BC: a behavior cloning agent which minimizes the negative log likelihood (NLL)  loss.
> - DQN: a naive offline DQN agent.
> - CQL: a discrete version of CQL algorithm.
> - DQN+BC: adding behavior clonging loss to the DQN agent, which is inspired by TD3+BC. Here, we use a softmax function to convert Q(s, a) to a distribution over actions, and then add the NLL loss to the origin TD loss.
>
> In the first case study, we trained these offline agents in the BreakoutNoFrameskip-v4 environment, where the offline dataset (2M) is collected by different checkpoints of an online DQN agent trained with 10M frames. We summarize the results in the following table:
>
> |                   | Offline DQN | CQL   | BC    | DQN+BC |
> | ----------------- | ----------- | ----- | ----- | ------ |
> | Reward            | 78.7        | 150   | 17.7  | 176.8  |
> | Optimal actions   | 25.7%       | 42.7% | 38.8% | 41.4%  |
> | Effective dim     | 10320       | 4047  | 1238  | 4461   |
> | Cosine similarity | 0.99        | 0.99  | 0.64  | 0.99   |
> | Dot product       | 994         | 5065  | 71    | 3687   |
>
> Here, we use the online DQN agent to approximate the optimal policy/value functions. The second line corresponds to the percentage of how often does the offline agent takes the same action as the online agent on a validation dataset (collected by online DQN checkpoints). We also list some representation metrics in line 3-5. We use the output of the penultimate layer of the QNetwork as the state representation.
>
> Our observations are consistent with some prior works [1, 2]. For example, TD-learning based RL algorithms usually learn representations that have large cosine similarity for consecutive states [1].  The representaitons of a performant TD-learning based agent usually relatively smaller effective dimensions [2]. On the other hand, such representation metrics seem to only hold for Q-learning based algorithms. For a behavior cloning agent, it learned representations with much smaller effective dimensions and the representations of consecutive states are less similar to each other. For the MinAtar game, we train the offline agents on the freeway environment. We find this toy environment is very simple that almost all the offline agents are able to solve this task.
>
> Overall, the observations from the experiments indicate that we need different metrics to evaluate representation qualities in image-based control tasks or state-based control tasks. In the paper, we focused on the state-based control tasks, and showed that we might need novel metrics to define what is good representations for such tasks.
>
> [1] Kumar, Aviral, et al. "Dr3: Value-based deep reinforcement learning requires explicit regularization." *arXiv preprint arXiv:2112.04716* (2021).
>
> [2] Lan, Charline Le, et al. "On the Generalization of Representations in Reinforcement Learning." *arXiv preprint arXiv:2203.00543* (2022).
>
> > **Q2:** description of the policy ranking metric
>
> Yes it is. In experiment, the policy ranking metric is computed as follows:
> - Collect a validation dataset $\{(s_1, s_2, \cdots, s_N) \}$ using a behavior policy.
> - For the i-th state $s_i$, we use 5 baseline agents to sample an action $\{(a_{i}^1, a_{i}^2, \cdots, a_{i}^5) \}$, where $a_i^1$ is the action sampled by the first baseline agent.
> - We use the online agent to evaluate the five $(s,a)$ pairs to approximate the optimal value function $(Q^*(s_i, a_i^1), Q^*(s_i, a_i^2), \cdots, Q^*(s_i, a_i^5))$
> - Then we sort the five $Q^*(s, a)$ values. If $Q^*(s_i, a^2_i)$ has the largest/smallest value, then we think the second agent has the best/worst policy on state $s_i$.
> - We report percentage of each agent that performs best/worst in each dataset.
>
> > **Q3:** lack of theoretical analysis
>
> Thanks for the suggestion. We admit that this paper is mainly an empirical work which attempts to analyze and compare the behaviors of some SOTA offline RL agents. We are looking forward to extend this work by introduing more theoretical analysis in the future.

---

### Official Review · Reviewer_LgJQ · 2022-07-10

**Rating:** 7
**Confidence:** 4
**Soundness:** 3 good
**Presentation:** 3 good
**Contribution:** 3 good

**Summary:**

The paper provides a comprehensive analysis of the state-of-the-art offline reinforcement learning algorithms. In particular, the paper evaluates the critics and policies from offline RL algorithms using several metrics, including representation probing, when the learned representations are used to predict various quantities, effective rank, action ranking, etc. The authors observe a mismatch between the quality of critics' predictions and the performance of the policy for some methods. Based on this insight, the authors propose a modification of IQL that achieves significant improvements over the original version.

**Questions:**

* Does the analysis help to improve the other methods as well?
* Does the analysis extends to other D4RL datasets? In particular, how have you trying running the metrics on antmaze tasks?

**Limitations:**

The paper can be improved by considering a wider variety of datasets from D4RL and applying the same analysis to enhance the other methods.

**Strengths And Weaknesses:**

# Strengths
* The paper is well written and easy to follow.
* I believe that the analysis of offline reinforcement learning methods is very valuable to the community and has received only limited attention so far.
* The analysis covers a variety of metrics for evaluating the critic and policy separately.
* The metrics introduced in the paper can help tune the components of offline reinforcement learning methods separately. The analysis has been applied in practice to the improvement of IQL.

# Weaknesses
* The paper focuses only on a subset of tasks considered in CQL and IQL.
* The paper focuses only on improving IQL.

To sum up, I believe this paper is relevant to the offline reinforcement learning community, and the pros outweigh the cons. Therefore, I recommend this paper for acceptance.

---

> ### Author Response · Authors · 2022-08-02
> **Response to Reviewer LgJQ**
>
> Many thanks for the thoughtful and detailed reviews. We are happy to hear that Reviewer LgJQ  agrees that our work addresses an important problem which only received limited attention in the community. We respond to specific questions and comments below.
>
> > **Q1:** only focus on a subset of tasks in CQL and IQL & antmaze tasks
>
> Thanks for the suggestion. Besides new experiments on the D4RL random datasets (Q2 Reviewer HrJL), we further add experiments in the D4RL antmaze tasks. It is notable that the antmaze task is a goal-reaching task, where we find it difficult to learn a near-optimal policy in this environment. Moreover, it is also challenging to define an optimal action in such tasks, where different action sequences are sometimes equivalent, i.e., "up + left" and "left + up". In addition, we also find it difficult to train stable COMBO agents in this task due to large model errors. The following table shows the results of the four model-free methods.
>
> |                                | TD3+BC | CQL  | IQL  | RIQL |
> | ------------------------------ | ------ | ---- | ---- | ---- |
> | Antmaze-umaze-v0               | 100.0  | 80.0 | 89.8 | 92.0 |
> | Antmaze-umaze-diverse-v0       | 71.4   | 62.5 | 68.4 | 67.5 |
> | Antmaze-medium-play-v0         | 0.0    | 65.4 | 71.4 | 66.8 |
> | Antmaze-medium-diverse-play-v0 | 0.0    | 62.5 | 74.4 | 67.5 |
> | Antmaze-large-play-v0          | 0.0    | 21.0 | 39.0 | 36.0 |
> | Antmaze-large-diverse-v0       | 0.0    | 11.8 | 47.2 | 42.8 |
>
> We can observe that the proposed RIQL algorithm performs slightly worse than the IQL baseline. This indicates that the relaxed policy constraint doesn't help in this task, where the agent needs to learn to stitch suboptimal trajectories. Since we do not have a near optimal policy/value functions at hand, we fail to have a close look at the details of the learned offline policy/value functions. In terms of the representation learning experiments, we observe similar results as in the locomotion tasks, where the inputs are also low-dimensional states.
>
> > **Q2:** only improves IQL
>
> In this work, we propose a series of new experiment setups to study the behavior of existing offline RL agents. From the evaluation results, we sometimes can diagnose the bottleneck of the existing algorithms. However, it is usually easier to find the bottleneck than to find a practical solution to improve the baseline. For example, the CQL algorithm only adds a conservative penalty to the TD loss. If we want to improve CQL, then we might need to replace the original conservative penalty. We might be able to use offline meta-learning to auto-tune the penalty coefficient to improve the performance. But such implementation sometime might be more complex.
>
> In the experiment, we use IQL as a case study to show that the proposed experiment setups did help to find the bottleneck. Due to the simplicity of the origin IQL algorithm, it is intuitive and easy to figure out a possible improvement solution. In addition, since IQL is a strong SOTA baseline algorithm, therefore an improvement upon IQL would also be meaningful.

---

### Official Review · Reviewer_HrJL · 2022-07-10

**Rating:** 5
**Confidence:** 5
**Soundness:** 3 good
**Presentation:** 4 excellent
**Contribution:** 3 good

**Summary:**

The paper surveys recent offline RL algorithms and seeks to analyze different factors contributing to their performance. Novel evaluation protocols are designed to analyze their representation and behavior. Based on this analysis, the authors propose a well-motivated modification to IQL which achieves strong results on several D4RL datasets.

**Questions:**

- It would be nice to disentangle the different contributions of model-based RL algorithms, could the analysis of model-based algorithms be better served by using MOPO, a more minimal algorithm?
- On line 157, the assumption is that an online learned “optimal” Q* would appropriately assign higher Q values to better true performing actions. Would this be flawed due to approximation? We may expect the Q functions’ values on OOD actions to be inaccurate. Could there be further issues from using a TD3-learnt policies which is different from the SAC policy used in many algorithms and used to generate the data?


**Limitations:**

The paper provides good insights on the datasets it tests, however:

- Limited evaluation on med/exp datasets, this could be expanded to include more D4RL-suite datasets.
- Incomplete information on seeds used during evaluation.
- Possible inaccuracy of using online Q-functions as ground-truth. Tabular settings where precise Q-function can be determined could elucidate this better.

**Strengths And Weaknesses:**

Strengths:
- Well-motivated range of analysis on the representation learned by an offline RL algorithm.
Proposed algorithms (RIQL, USS variants) show strong performance and are motivated by earlier findings in the paper.
- Novel and insightful analysis of how to integrate model-free methods into a model-based framework, and how to address the failures of a naive approach.

Weaknesses:
- On line 208 and 269, the uncertainty estimate used is the max mean-discrepancy, this measure is unusual as we would use the standard ensemble variance in supervised learning [1].
- Evaluation is solely on med/exp datasets, a good understanding of the strengths of each algorithm on mixed/random data where we may expect a higher extrapolation gap would be useful.

Minor:
- TD3BC should TD3+BC
- Typo on line 182: dose -> does
- Line 204, 262: the learned probabilistic model and model-based training are more accurately attributed to MOPO rather than COMBO.
In all tables and results: total random seeds should be included.

[1] Simple and Scalable Predictive Uncertainty Estimation using Deep Ensembles. Balaji Lakshminarayanan, Alexander Pritzel, Charles Blundell.

---

> ### Author Response · Authors · 2022-08-02
> **Response to Reviewer HrJL (Part 2)**
>
>
> > **Q4:** using TD3 as online policy
>
> In the experiment, we use checkpoints of the online policy (with different levels of performance) to collect validation datasets to approximate OOD samples. Since the offline datasets are collected by SAC agents, therefore, we want to use a different behavior policy (TD3) to collect samples that are dissimilar to the existing offline samples.
>
> To check if such selection would incur additional errors, we trained new online SAC agents in the same setting as TD3. We first compare the learned Q values of the TD3 and SAC by computing the symmetric mean absolute percentage error (sMAPE) $sMAPE(X, Y) = \frac 1 N \sum_{i}^N \frac{\vert X_i - Y_i \vert}{(\vert X_i\vert + \vert Y_i\vert)/2}$ on the validation set:
>
> |           | HalfCheetah | Hopper | Walker2d |
> | --------- | ----------- | ------ | -------- |
> | sMAPE (%) | 2.5%        | 7.0%   | 12.4%    |
>
> We can observe that near-optimal online TD3 and online SAC agents have similar Q-values on simple environments, i.e., HalfCheetah and Hopper. For more complex environments like Walker2d the sMAPE is higher, which is still less than 15%. In addition, we also rerun the experiments that use SAC as the online policy, and the observations are consistent with the previous results.
>
>
>
>
> > **Q5:** limitation of using online Q-functions & tabular experiment
>
> We admit that this is a weakness of our work, where using online Q-functions would incur approximation errors. However, in order to get a glimpse of the behaviours of RL agents in problems with continuous state/action spaces, such choices sometimes seem to be inevitable. One possible solution is to use the population-based RL [4], where we use an ensemble of different online Q-functions to mitigate the approximation errors.
>
> We also follow the reviewer's suggestion and run a case study on the tabular Bsuite Catch task [5], where a paddle can move right/left to catch a falling object in a 10x5 grid. We basically followed the experiment setup as in [6], where we first train an online DQN agent for 2000 episodes. Then we sample 10% transitions from the replay buffer as the offline datasets. In this tabular task, an object would fall randomly fall from the 5 positions in layer 1, and the paddle always starts at the center position on layer 10. We only receive a +1 score if the paddle successfully catches the object and a -1 score if the object falls on the ground. In such a tabular setting, we can easily compute the Q(s, a) functions for each state-action pair.
>
> In the experiment, the CQL agent can easily solve all 5 possible situations, while the offline DQN agent can only solve 3 situations. We then compared the learned Q-value functions of each agent and observed that both offline agents learn inaccurate Q(s, a) functions for OOD (s, a) pairs, where the DQN agent learns over-estimated Q-values for OOD samples while the CQL agent learns under-estimated Q-values. In such a simple toy task, the CQL's conservative penalty does help the agent to learn better Q-value functions that lead to the optimal policy. However, such observations can not easily generalize to problems with continuous state/action spaces. As we have seen in the experiments, sometimes such a conservative penalty would lead to over-conservative policies.
>
> [4] Parker-Holder, Jack, et al. "Effective diversity in population based reinforcement learning." *Advances in Neural Information Processing Systems* 33 (2020): 18050-18062.
>
> [5] Osband, Ian, et al. "Behaviour suite for reinforcement learning." *arXiv preprint arXiv:1908.03568* (2019).
>
> [6] Gulcehre, Caglar, et al. "Regularized behavior value estimation." *arXiv preprint arXiv:2103.09575* (2021).

---

> > ### Comment · Reviewer_HrJL · 2022-08-06
> > **Thanks for the responses**
> >
> > Thank you to the reviewer for engaging with the questions, particularly for discussing the limitations in Q5.
> >
> > On Q1, I'm reassured that the performance of the algorithm is not tied to max-mean. The fact that it performs similarly to ensemble std appears to be corroborated by [1] which also finds ensemble std to be the best uncertainty quantifier overall in model-based offline RL. This point could be useful to discuss.
> >
> > On Q2, I apologize if unclear, by mixed I meant medium-replay in D4RL. Having this data modality as well would bring the paper to the same standard of evaluation as other works in offline RL.
> >
> > On Q4, it's important to note that PBT *in general* doesn't necessarily provide a diverse set of value functions as PBT may be applied to any RL algorithm including those that don't learn value functions. Methods that create a diverse ensemble of value functions with the same hyperparameter setup include bootstrap Ensemble-DQN methods, etc.
> >
> > [1] Revisiting Design Choices in Offline Model Based Reinforcement Learning. Cong Lu, Philip Ball, Jack Parker-Holder, Michael Osborne, Stephen J. Roberts.

---

> > > ### Author Response · Authors · 2022-08-06
> > > **Thank you for the reply**
> > >
> > > > Experiment on the `medium-replay` dataset
> > >
> > > We did include the medium-replay dataset in the experiments, which is named as `-med-rep-v2`. We are sorry for the confusion, and we will clarify the datasets we used in the paper.
> > >
> > > > Problem of using PBT-based RL
> > >
> > > We agree that PBT-based RL doesn't necessarily provide a diverse set of value functions. From a practical perspective, using an ensemble of different value functions (learned with different random seeds/initializations) might be a reasonable and simple baseline to mitigate the problem of erroneous online value functions.

---

> ### Author Response · Authors · 2022-08-02
> **Response to Reviewer HrJL (Part 1)**
>
> We thank the reviewer for the insightful and detailed comments. We are glad to hear that Reviewer HrJL believes that our work provides well-motivated analysis for offline RL algorithms. We respond to specific questions and comments below.
>
> > **Q1:** using max min-discrepancy as uncertainty estimator.
>
> On line 208 and 269, we use the max mean-discrepancy as the uncertainty estimator, which is less popular in the supervised learning setting. The main reason we select it as the uncertainty estimator is that it has been used in some recently published related works, i.e., MOPO [1] and MoREL [2]. A more detailed discussion of the choice of uncertainty estimator in model-based offline RL can reference to Appendix C of [3].
>
> In our paper, the motivation of using an uncertainty estimator is to validate the assumption that -- noisy model-generated fake samples can be harmful to offline RL agents. Therefore, we choose to keep close to prior works. To compare the difference, we simply replace the max mean-discrepancy uncertainty estimator with the standard ensemble variance in the experiment of Table 11. We can observe a similar result that adding uncertainty-based sample selection (USS) helps to mitigate the negative effects of noise fake samples. Here, the result of `-USS2` corresponds to the use of standard ensemble variance as the uncertainty estimator.
>
> |       | IQL    | MIQL   | MIQL-USS2 | TD3+BC | MTD3+BC | MTD3+BC-USS2 |
> | ----- | ------ | ------ | --------- | ------ | ------- | ------------ |
> | Total | 690.04 | 334.16 | 696.78    | 676.82 | 448.99  | 703.18       |
>
> On the other hand, uncertainty quantification is an important open question in offline RL, which can help us to develop more robust offline RL agents. We leave it as an interesting direction for future work.
>
> [1] Yu, Tianhe, et al. "Mopo: Model-based offline policy optimization." *Advances in Neural Information Processing Systems* 33 (2020): 14129-14142.
>
> [2] Kidambi, Rahul, et al. "Morel: Model-based offline reinforcement learning." *Advances in neural information processing systems* 33 (2020): 21810-21823.
>
> [3] Yu, Tianhe, et al. "Combo: Conservative offline model-based policy optimization." *Advances in neural information processing systems* 34 (2021): 28954-28967.
>
> > **Q2:** no experiments on mixed/random data
>
> In Table 8 and Table 11, the `-med-exp-v2` task is a mixed dataset which contains both medium and expert samples. Thanks for the suggestion! We add experiments on the random dataset as follows:
>
> |                       | TD3+BC       | CQL          | COMBO        | IQL          | RIQL         |
> | --------------------- | ------------ | ------------ | ------------ | ------------ | ------------ |
> | HalfCheetah-random-v2 | 10.38 (1.07) | 14.46 (0.12) | 10.42 (8.30) | 11.77 (1.84) | 22.35 (3.45) |
> | Hopper-random-v2      | 8.56 (0.25)  | 7.36 (0.44)  | 0.69 (0.11)  | 8.09 (0.22)  | 9.19 (1.62)  |
> | Walker2d-random-v2    | 1.76 (1.05)  | 5.06 (0.48)  | 2.07 (1.38)  | 5.85 (0.64)  | 9.46 (3.66)  |
>
> As we can see, the proposed RIQL algorithm also achieves better performance than other baselines in other tasks.
>
> > **Q3:** clarification of using MOPO & random seeds.
>
> Thanks for the suggestions. We rewrite lines 204 and 262 to clarify the contribution of MOPO in the paper. Especially in Section 7, we update the writing to emphasize that the compared algorithms MIQL and MTD3+BC can be viewed as an application of MOPO and IQL/TD3+BC.
>
> In all exepriment, we report the mean and standard deviation over 5 random seeds.

---

### Author Response · Authors · 2022-08-02
**General answer to reviewers**

We'd like to thank all reviewers for their time and efforts on providing us with valuable feedback for this paper. We will address each reviewer's concerns in separate responses and offer a brief summary here.

We are very happy to see that reviewers generally agree that: (1) the work is well motivated [Reviewer HrJL]; (2) the proposed experiment setups/analyses are novel and helpful to better understand offline RL agents [Reviewer LgJQ, LmNq], which has only received limited attention in the community so far.

A common concern is that the experiments are only conducted on the d4rl mujoco tasks, which makes the current analysis limited to low-dimensional state-based locomotion tasks. To address this concern, we have run additional experiments and added more discussions. Below is a summary of what we have done:

- Add experiments to compare different uncertainty estimator [Reviewer HrJL].
- Add experiments on D4RL random tasks [Reviewer HrJL, LgJQ].
- Add experiments on D4RL AntMaze tasks [Reviewer HrJL, LgJQ, LmNq].
- Add case study on tabular task of BSuite Catch environment [Reviewer HrJL]
- Add case studies on image-based RL tasks [Reviewer LgJQ, LmNq].
- Comparing the difference of using TD3 and SAC as the online policy [Reviewer LgJQ]

Since some of the requested experiments are computationally intensive, which are still running. We will post an update in the discussion thread when the remaining results are done.

Also, we have corrected the typos and fixed the problem of duplicate references.

---

> ### Comment · Reviewer_LmNq · 2022-08-03
> **Edit: Found them**
>
> Edit: Nevermind, I was able to find the new results on the openreview comments (under other reviewers' comments).
>
> --
> Hi authors, thanks for your response.
>
> Quick question: I'm not able to find the additional experimental results in the main paper or appendix. Sorry if I missed it somewhere. Where can we view these results?

---

### Meta-Review · Area_Chair_Xom3 · 2022-08-20

**Recommendation:** Accept
**Confidence:** Less certain

**Metareview:**

The main strengths of this paper are that (1) it provides some interesting analysis that leads to some somewhat surprising findings, and (2) it presents and evaluates some new technical algorithmic ideas based on this analysis that lead to improved performance.

After the author discussion, the main weaknesses is that the new ant-maze results are somewhat disappointing, showing that the algorithmic ideas don't improve over IQL on a more complex problem setting. The ant maze tasks are a lot more interesting and complex than the standard locomotion tasks, and so this as a fairly major weakness. Of lesser importance, the title is not particularly descriptive, and could be used to describe a lot of papers. So, I would like to suggest to the authors to make the title more specific to the contributions of this paper.

Overall, the reviewers and AC think the strengths outweigh the weaknesses, especially since the analysis is interesting on its own and since there is some new analysis on more complex image-based settings, irrespective of the technical ideas only providing benefits on simplistic locomotion tasks. Nonetheless, we encourage the authors to use our feedback to further improve the paper.

**Award:**

No

---

### Decision · Program_Chairs · 2022-09-14

Accept